

# Intra-event evolution of elemental and ionic concentrations in wet deposition in an urban environment

Thomas Audoux[1], Benoit Laurent[1], Karine Desboeufs[1], Gael Noyalet[1], Franck Maisonneuve[2], Olivier Lauret[2,3], Servanne Chevaillier[2]

[1] Université Paris Cité and Univ Paris Est Creteil, CNRS, LISA, F-75013, Paris, France
[2] Univ Paris Est Creteil and Université Paris Cité, CNRS, LISA, F-94010, Créteil, France
[3] Univ Paris Est Creteil, CNRS, ENPC, Université Paris Cité, OSU-EFLUVE, F-94010, Créteil, France

*Correspondence to:* Thomas Audoux (thomas.audoux@lisa.ipsl.fr) ; Benoit Laurent (benoit.laurent@lisa.ipsl.fr)

**Abstract.**

Wet deposition is a crucial process that affects the lifetime of atmospheric particles by allowing them to be deposited by two different mechanisms, namely below the cloud and in the cloud. In order to estimate the mechanisms implied in the wet deposition, a measurement campaign was carried out in the Paris region to monitor the evolution of the chemical composition of wet deposition during rainfall events. Eight rain events have been sampled. The latter had different meteorological conditions, atmospheric dynamics and aerosol particles concentrations. Concomitant measurements of the chemical composition of aerosol particles and wet deposition allows calculating washout ratios from measurements taken at the beginning of the rainfall events, before the dilution effect occurred, and showed an increasing trend with increasing rainfall rates. The intra-event evolution of the chemical composition of wet deposition revealed the predominant role of meteorological parameters and local sources in the observed mass concentration variability. The contributions of in-cloud and below-cloud scavenging mechanisms were estimated for some rainfall events and found to vary depending on the specific sources, atmospheric dynamics and meteorological conditions. Overall, this study highlights the variability of wet deposition and its chemical composition, and the need to consider the specificities of each event to fully understand the underlying mechanisms.

## 1 Introduction

After emissions or formations and transport of aerosols in the atmosphere, wet deposition is one of the final sinks in their atmospheric cycle (e.g., Textor et al., 2006). Wet deposition involves two distinct mechanisms: in cloud scavenging (hereafter referred to as ICS) and below cloud scavenging (hereafter referred to as BCS). ICS refers to the scavenging of aerosols within the cloud, where they either act as condensation (or ice) nuclei or are captured by already formed droplets (Seinfeld and Pandis, 2016). BCS is the result of particles being captured through collision by raindrops as they fall via several size-related mechanisms (e.g., Brownian diffusion, interception, inertial impaction; Slinn, 1977). Through these two mechanisms, wet deposition includes locally emitted aerosols that can be scavenged from the atmosphere, as well as long-range transported aerosols that can be removed by precipitating cloud systems (e.g., Bertrand et al., 2008). Depending on the regions, wet deposition mechanisms collect atmospheric aerosols from different (natural, anthropogenic) sources that can be identified by their



chemical composition. By scavenging atmospheric pollutants and potentially toxic metals, wet deposition has an impact on air quality. Wet deposition of nitrogen (N), phosphorus (P) and trace metals can also serve as a significant input of nutrients species to terrestrial and marine ecosystems (e.g., Wright et al., 2018).

Worldwide observational measurement networks have shown strong spatial and temporal variability in the mass
and chemical flux of wet deposition (Vet et al., 2014; Keresztesi et al., 2019; Keene et al., 2015; Marticorena et al., 2017; Desboeufs et al., 2018). This variability can be observed at interannual, seasonal, daily, or intra-event scales and is dependent on the aerosol content, precipitation properties, and their interaction. Approaches based on only some of the measurable parameters have been used to document the scavenging of atmospheric particles by precipitation. One approach is to compute the washout ratio (hereafter referred to as WR), which is based on
the ratio of the mass or elemental concentrations of wet deposition to those of aerosols measured in the atmosphere (Chamberlain, 1960). WR is a parameter that integrates, without distinction of processes, the relative scavenging efficiency of particulate compounds and chemical elements by considering their transfer from air to water. WR has been regularly used to characterize wet deposition by precipitation for different types of particulate aerosols and chemical compounds found in various atmospheric environments (Jaffrezo et al., 1990; Duce et al., 1991;
Cerqueira et al., 2010; Marticorena et al., 2017).

The proportion of ICS and BCS in wet deposition is influenced by a number of factors, including the local environment (e.g., rural, urban) and associated emissions, meteorological variables such as rainfall amount, intensity and type, and aerosols content in the atmosphere such as its loading, their size and vertical distributions (Aikawa et al., 2014; Ge et al., 2016; Lim et al., 1991; Bertrand et al., 2008; Ge et al., 2021). The accuracy of the
representation of these mechanisms in global and regional modeling is still questionable (Croft et al., 2010), as there is insufficient data to constrain them accurately (Ryu and Min, 2022). Indeed, while BCS was considered to be less important than ICS regarding wet deposition in several modeling studies (Croft et al., 2010; Yang et al., 2015; Kim et al., 2021), recent observational studies have found that BCS represented a non-negligible fraction of the wet deposition (Xu et al., 2019; Ge et al., 2021; Chatterjee et al., 2010; Karşı et al., 2018; Audoux et al., 2023).
Grythe et al., (2017) also emphasized the significance of BCS, indicating that it is more responsible for the removal of aerosols in the lower atmosphere, while ICS dominates the wet removal in the free troposphere. These recent findings demonstrate the need to re-evaluate the importance of BCS in regional and global-scale modeling of atmospheric aerosols and thus the necessity to provide more in situ deposition measurements to better constrain them.

Several studies using sequential sampling have shown a decrease in concentration during the rain event, which is more pronounced in the first few millimeters of rainfall (e.g., Seymour and Stout, 1983; Jaffrezo et al., 1990; Aikawa and Hiraki, 2009). For example, Tanner et al., (2006) found that concentrations measured after 10 mm of rainfall can be 2 to 33 times lower than concentrations measured in the first 2 mm of rainfall, depending on the compounds studied. Sequential rainfall sampling allows the collection of successive rainfall fractions to monitor
the temporal variability of wet deposition (e.g., Laquer, 1990). It is of particular interest to study the dependence of wet deposition content on rainfall characteristics (intensity, droplet size and distribution), which also evolve during the event (Audoux et al., 2023). In addition, the study of the chemical composition of wet deposition and





its evolution throughout a rain event allows determining the influences of several aerosol sources (e.g., anthropogenic, natural). The intra-event evolution of rain chemical composition has also been used to discuss the relative contribution of ICS and BCS mechanisms to wet deposition (e.g., Aikawa and Hiraki, 2009; Ge et al., 2021; Audoux et al., 2023). Indeed, it is generally assumed that the first increments of the rain event are influenced by both mechanisms, while the last fractions could be attributed to ICS only (Aikawa and Hiraki, 2009; Chatterjee et al., 2010; Germer et al., 2007; Karşı et al., 2018; Desboeufs et al., 2010), although the relative proportion of ICS and BCS evolves during the event (e.g., Zou et al., 2022). Therefore, studying the evolution of wet deposition composition within a rainfall event provides valuable information on the temporal variability and the origin of scavenged aerosol particles, both in terms of sources of pollutant and BCS and ICS mechanisms.

A dedicated sequential precipitation sampler as well as conditioning and chemical analysis protocols were developed to document the intra-event variability of the dissolved and particulate chemical composition of rainfall (Audoux et al., 2023). The present study is based on the analysis of sequential rainfall sampling performed in late winter and spring 2022 at a study site in the Paris region, which included eight case studies with contrasting meteorological conditions, atmospheric loading, and chemical composition. It has three objectives: (1) to document the evolution of ionic and elemental composition of dissolved and particulate phase species in wet deposition with time and rainfall characteristics for contrasted rain events, (2) to discuss the parameters influencing the intra-event variability of wet deposition chemistry in terms of atmospheric aerosol particles sources, precipitation and meteorological properties and (3) to estimate the relative contribution of BCS and ICS mechanisms in the wet deposition.

## 2 Material and methods

### 2.1 Measurement site and sampling strategy of wet deposition and aerosol particles

The sampling site is located on the air quality station site operated by the Interuniversity Laboratory of Atmospheric Systems (LISA) and located at the University of Paris Est Creteil (UPEC) in the south-east of the Paris agglomeration (48.79 °N-2.44 °E) (Figure 1). The study site is in close proximity to various sources of pollution including nearby industries and an incinerator, highways, railway stations, and construction sites. Between July 2021 and July 2022, daily rainfall depths measured using a Précis-Mécanique rain gauge model 3 070 A (0.2 mm precision) at the study site ranged from 0.2 to 37.6 mm. 19% of rainy days presented rainfall depths lower than 0.4 mm, 12% were between 0.4 and 1 mm, 40% were between 1 and 5 mm, and 13% were higher than 10 mm. The sampling strategy is to investigate case studies sampled during an intensive measurement campaign during the winter and spring of 2022. During this period, the daily average $PM_{10}$ ($PM_{2.5}$) concentrations were around 17.5 (11.2) µg m$^{-3}$ with values reaching up to 57.5 (43.0) µg m$^{-3}$. Wet deposition collection is performed with a sequential sampler specifically developed at the LISA (Figure 1 A). Sampling, conditioning and analysis of rain samples are described in detail in Audoux et al. (2023), and thus, it is briefly reminded here.

The rain is collected using a Teflon pyramid funnel with a collection surface of 1 m² in combination with a sampling unit. This unit enables the automatic collection of up to 24 consecutive fractions of rain, adjustable from 0.05 to 2.0 mm, to study dissolved and particulate phase of the wet deposition. The sampling is conducted based





on volume, and as a result, it is dependent on the rainfall rate. The sequential sampler is able to correctly collect

rainfall fractions for low rainfall intensities, as well as for more intense rainfall recorded by the rain gauge and disdrometer, in comparison with standardized measurements (Audoux et al., 2023). The materials that make up the sampler have been chosen to allow analysis of the ionic and elemental composition of the dissolved and particulate phase at high and low concentration levels (from several mg L$^{-1}$ to hundreds of ng L$^{-1}$). The sampling bottles and materials that came in contact with the samples underwent a thorough washing protocol in clean-room

laboratory with ISO 7 and ISO 5 level controls.

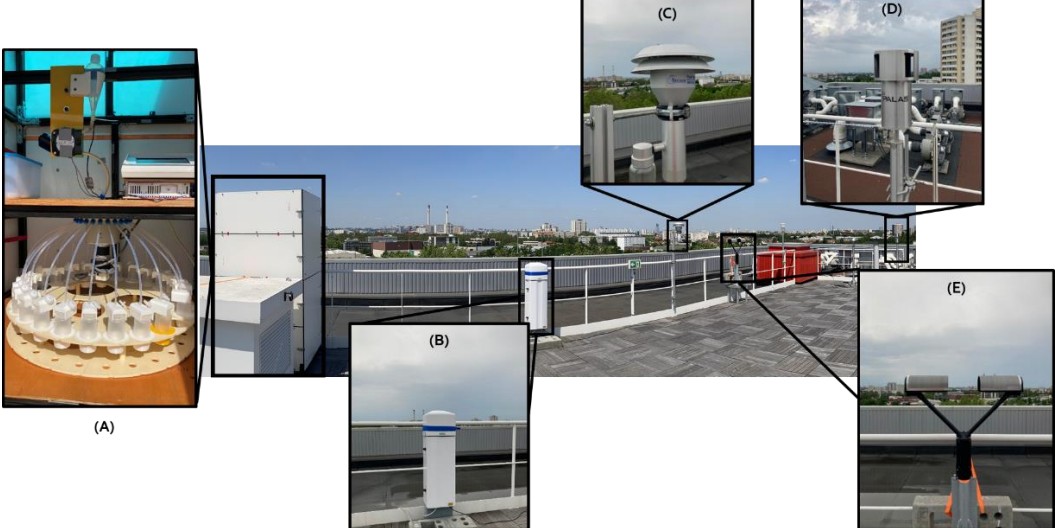

**Figure 1: Study site. (A) Rain sequential sampler; (B) Ceilometer; (C) PM$_{10}$ inlet for filter sampling; (D) FIDAS; (E) Disdrometer**

A summer rain event was collected in July 2021 (R1) and a winter rain event were collected in February 2022 (R2)

(Audoux et al., 2023). These case studies are completed here with 6 additional events collected in late winter and spring 2022, between March (R3, R4, R5, R6 and R7 rain events) and April (R8 rain event). For the 8 rain events studied, between 11 and 32 consecutive fractions have been sampled, the latter being collected within 10 seconds to 2 hours, depending on the rainfall rate.

Concomitant measurements on atmospheric aerosols and meteorological parameters during the rain sampling is

important for a more in-depth understanding of wet deposition mechanisms. Therefore, PM$_{2.5}$ and PM$_{10}$ aerosol mass concentration, as well as the particle size distribution (PSD) between 0.18 and 18 µm, are measured using the FIDAS (Figure 1 D), equipped with a TSP Sigma 2 inlet, with a 1 min time step. The FIDAS is an instrument used for regulatory air quality measurement of PM$_{2.5}$ and PM$_{10}$ mass concentration (LCSQA, 2021). Moreover, PM$_{10}$ aerosol particles are sampled on polycarbonate membranes (Nuclepore®, 0.4 µm porosity) using a PM$_{10}$

head sampling (Figure 1 C). Air sampling is done between 15 hours and 24 hours before the start of the rain and is stopped at the beginning of the latter. This allows characterizing the chemical composition of the atmospheric aerosol prior to rainfall. Rainfall rate and droplet size distribution (DSD) are measured using an OTT PARSIVEL®



(PARticle Size and VELocity, Figure 1 E) optical disdrometer with a time resolution of 30 seconds. In parallel, wind direction, wind speed as well as air temperature and relative humidity are measured using instrumentation

from Campbell Scientific© and are recorded with a time step of 1 min. The cloud base height and homogeneity of atmospheric column are measured using a ceilometer (Vaisala CL31, Figure 1 B). Parsivel disdrometers and ceilometers are typically used in multiple measurement networks for precipitation and cloud base height characterization (e.g., Haeffelin et al., 2005; Tapiador et al., 2010). FIDAS, ceilometer and disdrometer measurements are made continuously at the study site, while aerosols filter sampling and deposition measurements

are made on alert before or during rain events. This makes it necessary to regular follow-up the precipitation alerts.

**2.2 Elemental and ionic composition analysis of wet deposition samples and aerosol membranes**

After sampling, rain samples are quickly processed for ionic and elemental analysis, usually within a time frame of 1 to 12 hours after the end of rainfall. If immediate processing is not feasible, the samples are kept in a cool and dark environment at 6° C, and are processed within 24 to 48 hours. Treatment, filtration, and conditioning are done

in a clean-room laboratory with ISO 6 level controls, under a laminar flow hood (U15 filter) which is estimated to be equivalent to ISO 3. A pH meter (METTLER TOLEDO Seven2Go) is used to measure the pH of each sequential sample. Samples are then filtered through pre-cleaned Nuclepore® polycarbonate membranes with a porosity of 0.2 μm to separate the particulate phase from the dissolved phase. Following Audoux et al. (2023), the dissolved phase is then divided into two fractions. The first fraction (10 mL aliquot) is frozen until the analysis of water-

soluble major inorganic cations ($Na^+$, $NH_4^+$, $K^+$, $Mg^{2+}$, $Ca^{2+}$), anions ($Cl^-$, $NO_3^-$, $PO_4^{3-}$, $SO_4^{2-}$) and organic ions ($HCOO^-$, $CH_3COO^-$, $C_2H_5COO^-$, MSA, $C_2O_4^{2-}$) by Ion Chromatography (Compact IC Flex, Metrohm®, PRAMMICS Platform). The second fraction (two 15 mL aliquots) is acidified to pH = 1 with nitric acid (Suprapur®) and stored at 6° C until analysis of water-soluble Al, Ba, Ca, Cr, Fe, K, Mg, Mn, Na, Ni, P, S, Si, Ti and Zn by Inductively Coupled Plasma Atomic Emission Spectrometer (ICP-AES, Spectro ARCOS Ametek®).

The membranes are dried under laminar flow hood and conditioned for 48 h at a constant relative humidity of 45 – 50% and at T = 20 °C prior weighting using a precision microbalance (METTLER TOLEDO® XPR26C, PRAMMICS Platform). In order to accumulate a sufficient amount of material for analysis, several rain sequential samples can be filtered through the same filter. Conversely, when the particulate load is too high, rain fractions can be filtered through multiple membranes. Elemental composition (Al, Ba, Ca, Cr, Fe, K, Mg, Mn, Na, Ni, P, S,

Si, Ti, and Zn) of the particulate phase is determined using X-ray fluorescence spectrometer (XRF, ZETIUM 4 kW, Malvern Panalytical, PRAMMICS Platform). The 0.4μm porosity Nuclepore® membranes are also analyzed using XRF to characterize the elemental composition of the aerosol in the air prior to rainfall events. Our strategy is to monitor the elemental inorganic fraction of the aerosol in order to link it to the rainfall composition. It therefore allows us to characterize about 40% of the average aerosol composition in the Paris region (Airparif,

165 2021).

**2.3 Origin of scavenged aerosol particles**

The origin of scavenging aerosol particles can be discussed in relation with their chemical compositions and the trajectory of the air masses. We calculated enrichment factors (EFs, Taylor and McLennan, 1985) in order to determine the origin of elements found in the rain samples. Al is used as the reference of the Earth's crust (hereafter



referred to as $EF_X^{crust}$), and Na as reference of the sea salt (hereafter referred to as $EF_X^{sea\ salt}$). Equation 1 is used to calculate EF as follows:

$$EF_X(\%) = \frac{([X]/[ref])_{rain}}{([X]/[ref])_{crust\ or\ sea\ salt}} \times 100, \tag{1}$$

Where $([X]/[ref])_{rain}$ correspond to the ratio between the element X and the reference (Al or Na) concentrations in rainwater samples and $([X]/[ref])_{crust\ or\ sea\ salt}$ the concentrations in the continental crust or in the sea.

To complement local wind measurements at the study site, air mass trajectories were calculated using the HYSPLIT model (https://ready.arl.noaa.gov/HYSPLIT.php) (Draxler and Rolph, 2012). HYSPLIT is a retro-trajectory analysis used to study local to continental air mass dispersion and transport of atmospheric compounds, respectively (Celle-Jeanton et al., 2009; Bertrand et al., 2008; Calvo et al., 2012), and to determine the origin of air masses to identify sources of atmospheric substances, e.g., mineral dust, sea salt or anthropogenic (Vincent et al., 2016; Anil et al., 2017). Here, 48 h or 120 h depending on the event, backward trajectories were computed by the HYSPLIT model with GFS (0.25 °, global) from the study site (47.79 °N - 2.44 °W) at the surface (0 m a.g.l.) and at the cloud base height measured by the ceilometer.

**2.4 Determination of washout ratios and scavenging mechanism contributions**

WR is a parameter used to quantify the relative scavenging efficiency of particulate compounds and chemical elements by rain. It is based on the principle of a transfer of the compounds from the air to the water. Therefore, WR are determined from the ratio of the elemental concentration measured in the wet deposition ($C_{rain}$) to those measured in the aerosol filter ($C_{air}$) (equation 2).

$$WR = \frac{C_{rain}\ (\mu g\ kg^{-1})}{C_{air}\ (\mu g\ m^{-3})} \times \rho_{air}\ (kg\ m^{-3}), \tag{2}$$

WR make it possible to study the relative importance of some of the parameters involved in the mechanism of the transfer between the phases, such as rainfall rates (González and Aristizábal, 2012) or aerosol particle size (Jaffrezo and Colin, 1988; Cheng et al., 2021). As opposed to what is done in the literature, i.e., the calculation of the WR taking into account the whole event (e.g., Cheng et al., 2021), the sequential collection makes it possible to avoid being affected by the dilution effect reported in the literature (e.g., Jaffrezo et al., 1990) and to use atmospheric concentrations that are more representative of the scavenged aerosol. Indeed, we calculate the WR using equation 2, but, instead of using the mean elemental concentration of the rain event, we use the concentration measured in the first fraction of the rainfall, that is, the one mainly controlled by the BCS.

In order to discuss the relationship between aerosol and wet deposition, information is needed on both aerosol and rain, which we have for R2, R3, R5 and R8. In order to be able to calculate the WR, the homogeneity of the atmospheric column is a parameter to be taken into account in order to justify the representativeness of the aerosol measurements at the surface. Therefore, we will discuss the WR of the element only for R2, R3 and R8.





Determination of the ICS mechanism relative contribution to the measured wet deposition is determined using the mass concentrations of chemical species measured at the end of the rainfall ($C_{ICS}$), a period for which there is a steady state between BCS and ICS dominated by ICS, thus BCS is considered to be negligible (e.g., Aikawa and Hiraki, 2009). Different approaches are used to determine $C_{ICS}$, such as measuring after a certain amount of rainfall

(e.g., 5 mm; Aikawa and Hiraki, 2009; Xu et al., 2017) or selecting the lowest values during rainfall events (Karşı et al., 2018; Berberler et al., 2022). Some authors also fit an exponential decay law and use the constant value as $C_{ICS}$ (Ge et al., 2021), while others determine $C_{ICS}$ using the average value obtained during periods of lower mass concentration variations (Chatterjee et al., 2010). We determine $C_{ICS}$, using the VWM of the last fraction of rain, once a constant level is reached. The wet deposition flux due to the ICS mechanism can thus be calculated using

$C_{ICS}$ and $P_{tot}$, the total rainfall depth of the rainfall (equation 3) as done previously in the literature (Xu et al., 2017; Aikawa et al., 2014; Ge et al., 2021).

$$F_{ICS} = C_{ICS} \times P_{tot}, \tag{3}$$

Then, the wet deposition flux due to BCS mechanism ($F_{BCS}$) is determined by subtracting $F_{ICS}$ from the total (dissolved + particulate) wet deposition ($F_{total}$). Relative contributions of BCS ($BCS_C$) and ICS ($ICS_C$) to wet

deposition can be obtained using equations 4 and 5, respectively.

$$BCS_C = \frac{F_{BCS}}{F_{total}}, \tag{4}$$

$$ICS_C = \frac{F_{ICS}}{F_{total}}, \tag{5}$$

The calculation of $ICS_C$ requires to be in situations where the mass concentrations obtained at the end of the rain event can be considered representative of the concentrations of droplets in the cloud. To ensure this, we selected

rainfall events for which the measurements indicated an effective scavenging of the atmospheric column, with a predominant relative contribution of ICS at the end of the event. To select these events, we used the following criteria: 1) the decrease of concentrations measured in the wet deposition, reflecting the evolution of the contribution of the BCS; 2) the decrease of atmospheric concentrations measured using the FIDAS, suggesting a progressive scavenging of the air column under the cloud; and 3) constant concentrations of wet deposition at the

end of the event, indicating a steady state between ICS and BCS. Thus, the evolution of the concentrations in the wet deposition, associated with the evolution of the atmospheric concentrations, makes it possible to discuss the relative contributions of the scavenging mechanisms for R1, R2, R4 and R8 case studies.

## 3. Results

### 3.1 Description of wet deposition case studies

Eight rainfalls constitute a data set illustrating various cases in terms of aerosol concentrations and compositions as well as precipitation properties. The properties of the 8 rainfall events studied are listed in Table 1. The rainfall events are characterized by variable rainfall depths ranging from 0.9 to 6.9 mm and mean rainfall rate from 0.4 mm h$^{-1}$ to 11.5 mm h$^{-1}$. Depending on the rainfall depths and rates, the sampling resolution was adapted. For





example, R7 was collected in 22 fractions of volumes ranging from 80 to 440 mL for a rainfall depth of 3.04 mm over 30 min, while R8 was collected in 32 fractions of volumes ranging from 60 to 820 mL for a rainfall depth of 6.9 mm and lasted several hours. Note that for R7, the sampling setup allowed us to only collect the first 3.04 mm of rain of the total event (10.3 mm). Our dataset consists of one (12.5%) event with a rainfall depth of less than 1 mm, one (12.5%) with a rainfall depth of more than 5 mm and the other (75%) representing rainfall depths between 1 and 5 mm.

Atmospheric aerosol mass concentrations at the beginning (average over the 30 min prior to the onset of the rainfall) of R1, R2, R6 and R7 events are primarily controlled by $PM_{2.5}$, which represents 63–84% of $PM_{10}$ concentrations. R3 is characterized by a lower proportion of $PM_{2.5}$, which represents 38% of $PM_{10}$, while $PM_{2.5}$ measured for R4, R5 and R8 correspond to 46–53% of $PM_{10}$. R1 to R4 took place on days with low particle concentrations, with $PM_{10}$ concentrations lower than of 20 $\mu g$ $m^{-3}$. During these events, rain had the effect of

reducing atmospheric $PM_{10}$ concentrations by 11–53% (Table 1). However, this illustrates the overall effect of the rain event without taking into account the increases in air concentrations that may have been observed during the events (e.g., R8). On the other hand, R5 to R7 occurred on days marked by high concentrations of both $PM_{2.5}$ (33–40 $\mu g$ $m^{-3}$) and $PM_{10}$ (47–63 $\mu g$ $m^{-3}$). The latter took place not only during a typical spring pollution episode (Favez et al., 2021), but also during a mineral dust intrusion from North Africa, as shown by a multi-model dust optical

depth simulation provided by the WMO Barcelona Dust Regional Centre (Supplement S1, https://dust.aemet.es, Basart et al., 2019). During these events, rain was less effective at reducing $PM_{10}$ concentrations. While R5 is characterized by a decrease in the $PM_{10}$ concentration of the order of 17%, R6 and R7 show no variation of an increase in the $PM_{10}$ concentration (Table 1). Even though R8 occurred on a day with low particle concentrations, this event was also marked by the intrusion of mineral dust from northern Africa (Supplement S1, Table 1).

Total wet deposition fluxes are ranging from 11 to 107 g $m^{-2}$, and do not seem to be correlated with rainfall depth nor rainfall rate (Table 1). Although wet deposition fluxes are higher for events characterized by higher surface $PM_{10}$ mass concentrations (R5, R6 and R7), they still show a factor of 4 between events characterized by a similar surface $PM_{10}$ mass concentration (R1, R2, R3, R4 and R8).

**Table 1. General information of studied rainfall events.**

| Period Date. Hour (UTC) | Rain | Number of rain fractions | Mean rainfall depth (mm) | Mean rainfall rate (mm h⁻¹) | Pre-rain $PM_{10}$ concentration ($\mu g$ $m^{-3}$) | $PM_{2.5}/PM_{10}$ fraction (%) | After-rain $PM_{10}$ concentration ($\mu g$ $m^{-3}$) | Origin of air masses | Cloud base height (m) | Total wet deposition fluxes (g $m^{-2}$) | pH range |
|---|---|---|---|---|---|---|---|---|---|---|---|
| Jun.27 6:55 – 12:25 | R1 | 21 | 2.65 | 0.48 | 18.7 | 61 | 12.2 | West | - | 11.3 | 5.0 – 6.0 |
| Feb.10 17:28 – 20:55 | R2 | 17 | 1.33 | 0.49 | 13.0 | 70 | 6.1 | West | 500 – 1 000 | 12.0 | 6.3 – 6.8 |
| Mar.11 11:06 – 13:19 | R3 | 15 | 1.03 | 0.69 | 11.8 | 36 | 9.0 | South-South West | 2 000 | 25.7 | 7.8 – 7.1 |
| Mar.11 14:16 – 17:24 | R4 | 17 | 4.42 | 1.41 | 9.9 | 55 | 4.6 | South | 1 200 – 1 500 | 27.8 | 6.0 – 6.9 |
| Mar.29 13:10 – 16:50 | R5 | 11 | 0.90 | 0.40 | 62.6 | 52 | 52.1 | North-East (surface) South (cloud base) | 1 500 – 2 000 | 106.3 | 7.8 – 8.4 |



| Mar.30 4:55 − 9:31 | R6 | 15 | 1.20 | 0.43 | 44.3 | 82 | 51.7 | South | 200 | 107.1 | 7.2 − 8.0 |
| Mar.30 15:32 − 16:01 | R7 | 22 | 3.04 | 11.5 | 47.2 | 80 | 46.6 | South | 1 000 | 69.0 | 6.6 − 7.4 |
| Apr.13 3:00 − 12 :12 | R8 | 32 | 6.94 | 0.90 | 11.9 | 46 | 10.6 | South | 200 − 2 000 | 47.9 | 6.1 − 7.1 |

According to the HYSPLIT 48 h back trajectory calculation, the origin of the air masses scavenged at the study site remained constant during the duration of the rain events, except for R6 and R8 (Figure 2). The air masses for R1 and R2 came from the Atlantic Ocean. R3 and R4 had air masses from the Mediterranean at the surface and from Spain and Portugal at the cloud base. For the other events, influenced by mineral dust intrusion from North Africa, the calculation of HYSPLIT back trajectories has been performed over 120 hours with the same conditions.

For R5 and R6, the air masses at the surface came from the United Kingdom via the North Sea and Germany, while the air masses at the cloud base came from North Africa (south of Tunisia/west of Libya) for R5 and from the Mediterranean Basin and Italy for R6. In the second phase of the event R6 (after 9:00 UTC), the air masses at the surface also came from the Mediterranean Basin. For R7, the air masses at the cloud base came from the Mediterranean Basin and the air masses at the surface came from Libya. For R8, the beginning of the event was

characterized by air masses coming from the Atlantic through North Morocco and Spain at the cloud base and from northern Tunisia at the surface. During the event, the origin of air masses evolved and came from different places in Northern Africa (Morocco, Algeria, and Tunisia) depending on altitude. This analysis of the back trajectories shows a close alignment between the origins of these large-scale air masses and the wind directions measured at the surface at the instrumented site in Creteil.

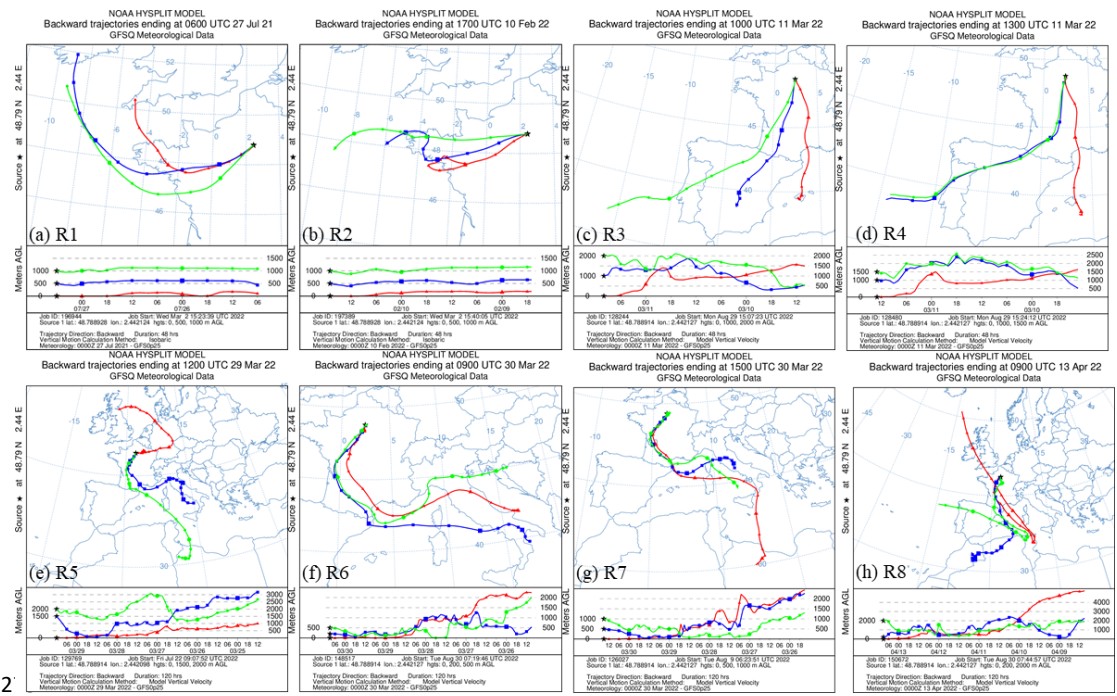



**Figure 2. HYSPLIT 48-h (a-d) and 120-h (e-h) back trajectories of surface and cloud-base air masses for the collected rain events.**

Figure 3 shows the cumulative drop size distribution (DSD) and droplets fall velocity for each rain event. The droplet fall velocity curve, plotted against droplet size, closely resembles the Gunn and Kinzer curve commonly observed for droplets in the range 0.05 to 29 mm (Gunn and Kinzer, 1949). Recorded droplets fall velocity ranges are similar across all events, ranging from 0.15 to 8.8 m s$^{-1}$. However, there are differences in the number and size of droplets between events. R8, which has a greater rainfall depth, has a higher number of drops. On the other hand, R7 is associated with larger droplets ($D_{50}$ = 0.94 mm), which explains the higher rainfall rate compared to other events.

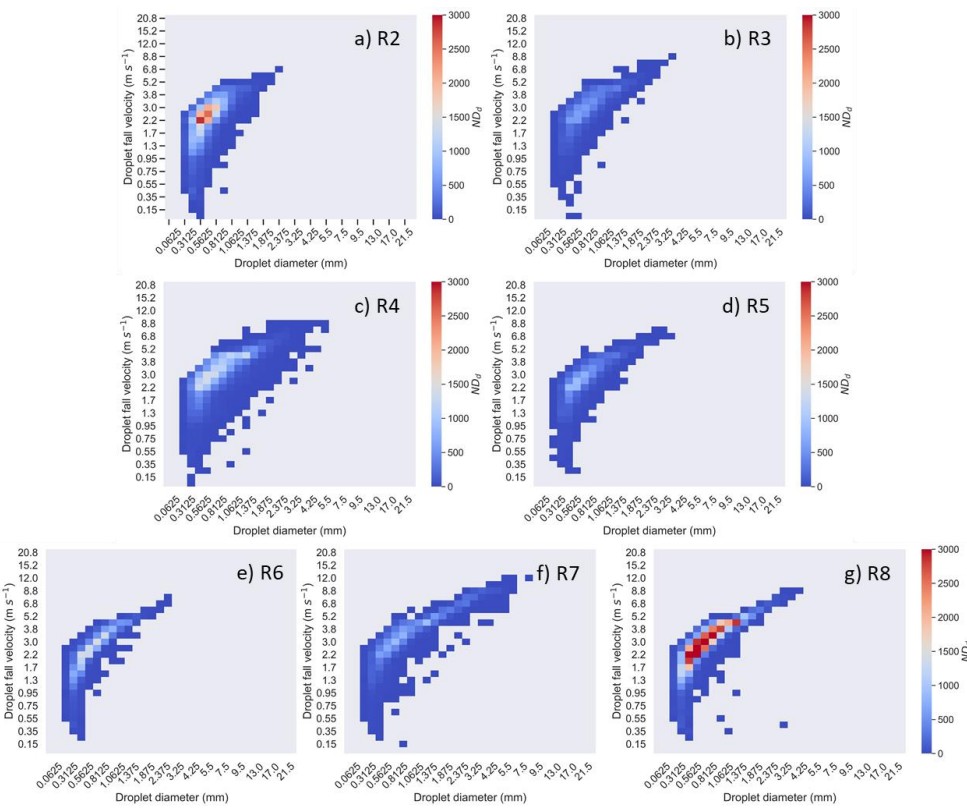

**Figure 3. Evolution of droplets fall velocity (m s$^{-1}$) and total number of droplets (ND$_d$) as a function of the droplet diameter (mm) for each rain events, except for R1 for which the disdrometer was not in operation.**

The precipitation observed by the ceilometer is shown by the yellow-orange to red signals (Figure 4), which shows the precipitation originating from the cloud base and reaching the surface. The signal intensity also illustrates the homogeneity of the atmospheric load on the air column from the surface to the cloud base, except for R5 and R8 where the air columns below the clouds appear inhomogeneous. For example, the measured signal intensity is lower after event R4 (light blue) than before the event (blue-green). This illustrates the effective leaching of the





atmospheric column, which is consistent with a decrease of the order of 53% in the PM$_{10}$ concentrations measured at the surface. In addition, for R5, one can also see an aerosol layer between 2000 and 3000 m altitude that

gradually get down to the cloud level just before it starts raining. Rain events have varying cloud base heights (from 200 m for R6 up to 2 000 m for R8) which, however, can fluctuate within the same event as seen for R8. By combining surface rainfall measurements (disdrometer, Figure 4) with air column measurement (ceilometer, Figure 5), it can be seen that decreases in rainfall rates are associated with decreases in signal intensity measured by the ceilometer in some cases, indicating altitude precipitation that did not reach the surface (e.g., R2 around

14:00, R8 around 05:00).

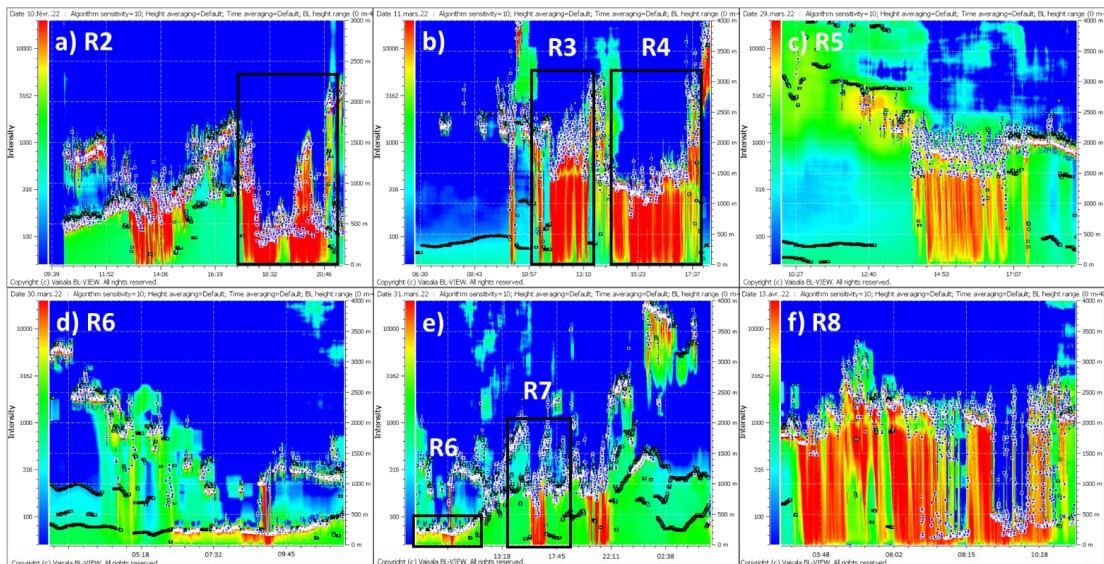

**Figure 4. Temporal evolution of the signal intensity measured by the ceilometer as a function of time and altitude. The blue and white squares represent the cloud layer base, the black and light blue squares represent the boundary layer height and the color scale corresponds to a proxy of the atmospheric aerosol load. The intense orange-yellow to red lines**

**correspond to the precipitating raindrops.**

The information collected makes it possible to describe 8 case studies, illustrating contrasting situations in terms of meteorological conditions, dynamics and atmospheric aerosols loads.

**3.2 Classification of case studies**

Differences observed in elemental and ionic mass concentration and composition of wet deposition led us to

classify the events. Volume weighted mean (VWM) mass concentrations of the particulate and dissolved phases for each rain event are represented in figure 5.

Regarding the particulate phase, the average concentrations of major elements were found to be higher for R5 and R6 (Al: 3 310 – 3 560 µg L$^{-1}$, Fe: 2 650 – 3 170 vs. µg L$^{-1}$, Si: 8 500 – 10 300 µg L$^{-1}$) by more than one order of magnitude in comparison to R1 to R4 (Al: 28 – 130 µg L$^{-1}$, Fe: 54 – 210 µg L$^{-1}$, Si:114 – 590 µg L$^{-1}$). Although

the mass concentrations of the major elements in the particulate phase of the rainfall vary within a factor of 85, the



particulate phase is primarily composed of Si, Fe, and Al, with a relative contribution ranging from 73 – 85% regardless of the event. The particulate Ca content, on the other hand, was found to vary between 3 – 12% depending on the rain event.

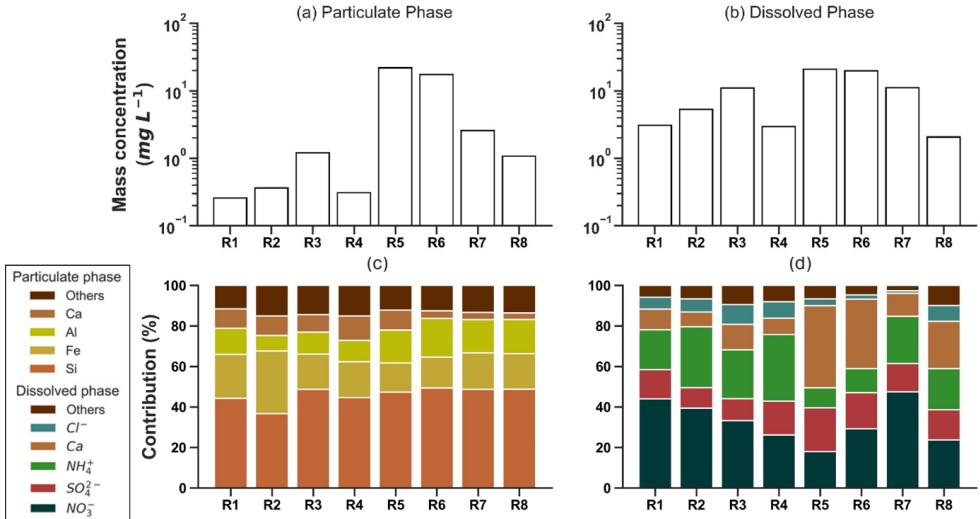

**Figure 5. Volume weighted mean mass concentrations (mg L$^{-1}$) of (a) particulate and (b) dissolved phases. Contribution of elements in the elemental composition of particulate phase (c) and of chemical species of ionic and elemental composition of dissolved phase (d).**

Regarding the dissolved phase, R4 and R8 are the rainfall events characterized by the lowest dissolved phase VWM concentrations (~ 2 to 3 mg L$^{-1}$) and the largest rainfall amounts (4.4 mm and 6.9 mm for R4 and R8, respectively). These results are consistent with the dependence of wet deposition concentrations with precipitation amount and the dilution effect documented in the literature (e.g., Jaffrezo et al., 1990). The largest concentrations are of the order of 21 mg L$^{-1}$ and correspond to the events marked by the mineral dust intrusion from northern Africa but also the lowest precipitation amounts (0.90 mm for R5 and 1.20 mm for R6). The latter are of the same order of magnitude as values found in the literature for semi-arid environments with values of the order of 20 mg L$^{-1}$ (Tuncel and Ungör, 1996; Tripathee et al., 2020) or for urban environments in Europe during mineral dust intrusion (11.9 – 20.6 mg L$^{-1}$; Avila et al., 1997; Rogora et al., 2004; Anatolaki and Tsitouridou, 2009), marked by high contents of calcium and species of crustal origin (e.g., Avila et al., 1997; Oduber et al., 2020). Dissolved Ca and SO$_4^{2-}$ contents of the same order of magnitude 20 – 41% by mass are also found (Rogora et al., 2004; Anatolaki and Tsitouridou, 2009). The rain events are not characterized by the same contents and relative proportions of acid (NO$_3^-$, SO$_4^{2-}$) or neutralizing (NH$_4^+$) species depending on the rainfall. The dissolved phase is mainly composed of SO$_4^{2-}$, NO$_3^-$ and NH$_4^+$ (SNA), between 58 and 85% by mass of the analyzed species for R1, R2, R3, R4, R7 and R8. In contrast, R5 and R6, and to a lesser extent R8, are composed of a non-negligible proportion of Ca in the dissolved phase (23 – 40%).



The variations in concentrations of not only acid species, but also neutralizing compounds, lead to different pH values in the rainfall (Table 1). The progressive scavenging of these compounds during the rainfall event also results in variations in pH (Asman et al., 1982). These variations in pH can be observed between different events. For instance, R1 has a lower pH (pH < 5.6) resulting from lower average concentrations of neutralizing species. Rains R2, R3, R4, R7 and R8 have higher pH values ranging from 6.2 - 6.8, and even basic for R5 and R6 (7.5 - 8.0). The basic nature of R5 and R6 rains is attributed to the higher Ca contents of mineral dusts present in these

rains, which is in agreement with the influence of dust intrusion, as previously described (Ma, 2006; Oduber et al., 2020).

To go further in the interpretation, EFs presented in table 2 as well as origin of air masses (Table 1), are used to classify case studies into three groups: (i) R1 to R4, characterized by air masses from the west and south of France and a significant enrichment in Ca (EF> 15), Ni (EF> 10, except R4), P (EF> 30, except R1) and very high for Zn (EF> 120) and S (EF> 1000); (ii) R5 and R6, characterized by a contribution of mineral dust and EFs reflecting

mineral sources signature (between 1 and 2), except for Zn (8.0 – 13) and S (119 – 136), which are still lower than the other rains; and (iii) R7 and R8, characterized by low EFs (<10) for all elements, but higher than R5 and R6 ones, except for Zn (26 – 44) and S (175 – 438).

**Table 2. Enrichment factors ($EF^{crust}$) of elements measured in the rain events relative to the upper continental crust.**
**Bold values indicate significant enrichment of the element ($EF^{crust}$> 10).**

| $EF^{crust}$ | Ba | Ca | Cr | Fe | K | Mg | Mn | Na | Ni | P | S | Sr | Ti | V | Zn |
|---|---|---|---|---|---|---|---|---|---|---|---|---|---|---|---|
| **R1** | 7.9 | **19** | 5.6 | 2.5 | 3.2 | 1.4 | 3.8 | 4.8 | **17** | 8.1 | **1 281** | 5.4 | 1.7 | 2.6 | **226** |
| **R2** | **20** | **31** | **16** | 6.3 | **12.7** | 1.0 | **9.3** | **16** | **52** | **53** | **1 853** | **9.9** | **4.2** | **16** | **396** |
| **R3** | 6.6 | **25** | 5.2 | 2.6 | 5.9 | 1.8 | 5.4 | **13** | **11** | **33** | **1 060** | 6.3 | 3.0 | 5.0 | **121** |
| **R4** | 7.5 | **17** | 5.6 | 2.7 | 7.0 | 1.5 | 5.2 | 9.9 | 5.4 | **38** | **1 521** | 5.2 | 3.0 | **14** | **190** |
| **R5** | 2.6 | 6.6 | 2.3 | 1.4 | 1.2 | 1.0 | 1.5 | 0.37 | 0.9 | 3.8 | **136** | 2.8 | 1.7 | 3.1 | **13** |
| **R6** | 2.1 | 4.4 | 1.8 | 1.3 | 1.1 | 0.92 | 1.2 | 0.25 | 2.2 | 1.4 | **119** | 2.0 | 1.5 | 2.6 | 8.0 |
| **R7** | 4.2 | 7.1 | 4.1 | 1.8 | 1.6 | 1.1 | 1.9 | 0.53 | 3.7 | 4.1 | **438** | 2.7 | 1.8 | 3.1 | **44** |
| **R8** | 3.2 | 6.2 | 2.9 | 1.7 | 1.7 | 1.1 | 1.8 | 1.6 | 5.1 | 3.2 | **176** | 2.1 | 1.8 | 3.3 | **26** |

These elements make it possible to illustrate marked differences in terms of chemical concentrations and composition for the wet deposition events. The chemical signature allows us to classify rain events into three categories: R1, R2, R3 and R4 show a marked anthropogenic signature and are hereafter referred to as "anthropogenic" events; R5 and R6 illustrate a distinct mineral dust signature and hereafter

referred to as "mineral dust" events; when R7 and R8 correspond to mixing conditions and are hereafter referred as "mineral dust-anthropogenic" events.

## 4. Discussion

### 4.1. From aerosol particles and wet deposition compositions to washout ratios

Total mass concentrations measured in the first fraction of rainfall events (0.06 to 0.10 mm) are higher when pre-
rain $PM_{10}$ surface concentrations are greater (Table 1). However, for R2, R3 and R8, $PM_{10}$ concentrations are of





the same order of magnitude (11.8 – 13 µg m$^{-3}$) while total mass concentrations in their first fraction differ by a factor 1.8 (R2: 28.1 mg L$^{-1}$; R3: 49.8 mg L$^{-1}$; R8: 38.7 mg L$^{-1}$). The latter is higher when the PM$_{2.5}$/PM$_{10}$ ratio is lower (Table 1). This suggests that PM$_{2.5}$ are scavenged less effectively than coarser particles (PM$_{2.5 – 10}$). R6 and R7 events are characterized by similar pre-rain PM$_{10}$ surface concentrations as well as similar PM$_{2.5}$/PM$_{10}$ ratios.

However, R6 event shows total mass concentrations in the first fraction 2.4 times higher than R7 (68.3 mg L$^{-1}$). This can be explained by the long-range transport of mineral dust and thus the influence of particles present at altitude. Therefore, wet deposition fluxes at the beginning of rainfall seem to be primarily correlated to PM$_{10}$ surface concentrations and secondly to the coarse fraction (PM$_{2.5 – 10}$ / PM$_{10}$). This is consistent with the aerosol size dependence of scavenging mechanisms and the minimal efficiency of the BCS mechanism between 0.2 and

2 µm (e.g., Greenfield, 1957).

A measurement strategy has been developed for the documentation of both aerosol and wet deposition. As described section 2.3, sampling of PM$_{10}$ aerosols have been done during several hours before the beginning of the rainfall event for R2, R3, R5 and R8. The aerosol elemental composition sampled before R2, R3, R5 and R8 events, are referred to as E2, E3, E5 and E8 respectively. We observe various levels of PM$_{10}$ mass concentration

measured with the FIDAS during the sampling of aerosols, ranging in average from 12 to 44 µg m$^{-3}$ depending on the rain event. E2 sampling air mass comes from east and north, E3 and E8 come from South-East and E5 come from North-East. These directions are consistent with the backward air mass trajectory calculated with HYSPLIT for the beginning of the rain event. Thus, these samples correspond to the background chemical composition of the study site prior to the rain events. XRF analysis coupled with determination of the elements oxides masses

allows characterizing 15 to 55% of the measured mass concentration, depending on the situations.

Figure 6 depicts the total concentration of elements (dissolved + particulate) in the first rain fraction (µg kg$^{-1}$), plotted against the total concentration of elements measured in the aerosol (µg kg$^{-1}$) for R2, R3, R5 and R8 rain events for which both information on the composition of atmospheric aerosols before the rain and of wet depositions were measured. These figures illustrate the evolution of the chemical composition of major and trace

elements between the atmospheric aerosols sampled at the surface and the first fraction of the wet deposition. As an example, for similar mass concentrations in the aerosol phase (0.02 µg kg$^{-1}$), we found higher P concentration in the first fraction of R2 (88 µg kg$^{-1}$) in comparison with Ba (14 µg kg$^{-1}$). Therefore, we can note that some elements seem to be enriched in the rainfall in comparison with the aerosol phase. This could be due to either an additional source of elements in the rain (e.g., ICS), a difference in BCS efficiency due to a higher scavenging

efficiency compared to other elements, or a contribution of PM with a diameter greater than 10 µm.

Figure 6 allows us to discuss the washout ratio (WR) for the chemical elements analyzed. The element concentrations range over several orders of magnitude leading to WR ranging from less than 2000 (corresponding to the dashed lines on Figure 8) to more than 10 000 (corresponding to the solid line).





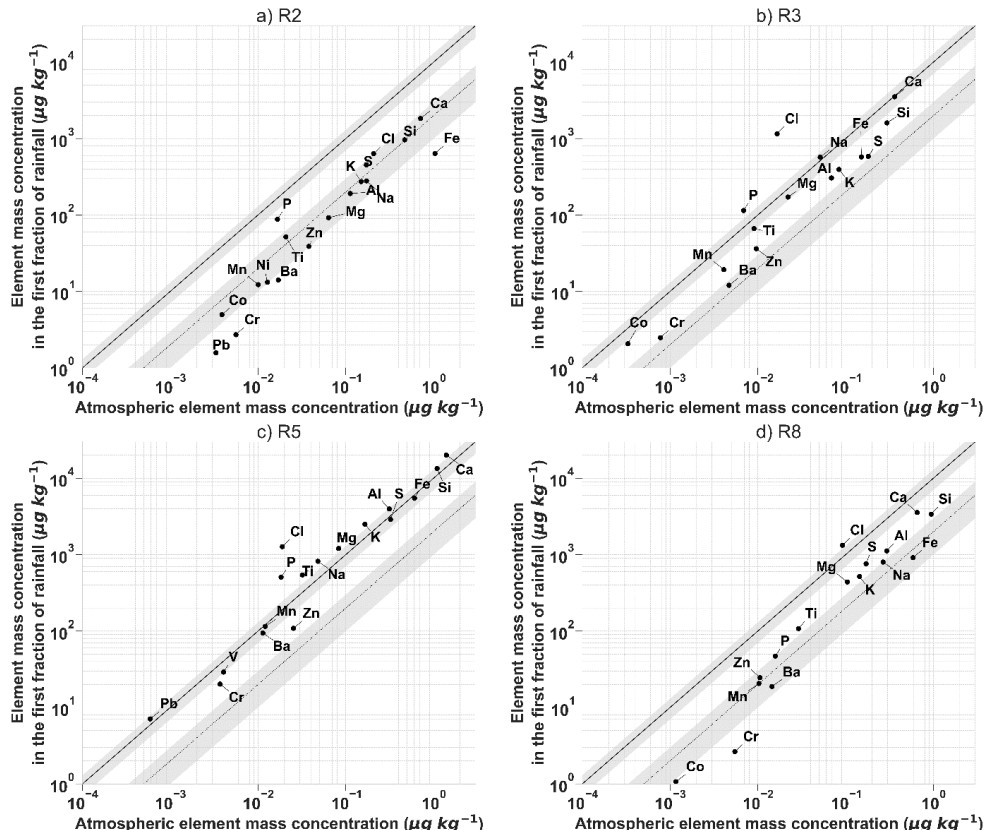

**Figure 6. Element mass concentration in the first fraction of the rainfall (µg kg⁻¹) as a function of the element mass concentration in the aerosol (%) of R2 (a), R3 (b), R5 (c) and R8 (d). The solid lines with envelopes correspond to washout ratios of the order of 10 000 ± 3 000, while the dashed lines with envelopes correspond to washout ratios of 2 000 ± 1 000.**

WR can be calculated from the information in Figure 6 (Supplement S2). Firstly, it is important to note that the values obtained from our calculation using the first instants of rain are at least in the upper range of the values reported in the literature, and even higher. This is due to the absence of the dilution effect in our case studies. In addition, the significant presence of aerosols at high altitudes observed with the ceilometer (Figure 5c) highlights the fact that the measured surface concentrations may not correlate with the amount of aerosols present in the atmospheric column. This explained higher level observed in the concentration of the first fraction of rainfall (Figure 6 c) and thus, WR are not calculated for R5.

Within the same event, WR values vary by a factor of 11 to 30 from an element to another (Figure 6). WR of elements found in R2 are primarily in the 2 000 ± 1 000 envelopes, while WR of R3 are systematically higher. Regarding R8 events, we observe an intermediate behavior in terms of WR values. These differences have been discussed in the literature as being due to differences in aerosol characteristics such as size, hygroscopicity or the additional contribution of gas phase scavenging (e.g., Jaffrezo and Colin, 1988; Cheng et al., 2021; Kasper-Giebl et al., 1999). Our results support the conclusions drawn in Cheng et al.'s (2021) literature review, as we found that





lower washout ratios (WRs) were calculated for species with higher $PM_{2.5}/PM_{10}$ ratios, except for S in rainfall events R2 and R8, where it may have acted as a condensation nucleus. Additionally, WRs appeared to vary based on the solubility of the chemical species, with higher WRs observed for Cl, Na, S, Ca compared to Cr and Co

(Figure 6, Supplement S2). Our findings, when considering the whole event for the WR calculation, also aligned with Jaffrezo and Colin's (1988) work, indicating that the concentration evolution within the same event can be attributed to a different scavenging efficiency depending on the element, partly due to its size (Supplement S3).

There is also a large variation between different events for the same chemical species. We found that for every element that, with the exception of S, this variability seems to be consistent with a decreasing trend in WR while

$PM_{2.5}/PM_{10}$ fraction increase (Table 1) as well as with an increasing trend in WR with increasing rainfall rates. Our results, which to the authors' knowledge are the first to be unaffected by a dilution effect, thus highlight the importance of rainfall rates and of fine fraction of aerosol particles on WR.

### 4.2. Intra-event evolution of wet deposition content

We were able to quantify for each event an overall decrease in mass concentrations of the particulate (up to a factor

of 50) and dissolved (up to a factor of 35) phases (Supplement S4). The decrease factors (DF), i.e., the ratio of the mass concentration of the first fraction to the last fraction of rainfall, were more pronounced for "anthropogenic" than for "mineral-dust" rainfall, consistent with the difference in terms of rainfall amount (1.0 – 4.4 mm vs. 0.9 – 1.2 mm), and more marked for the particulate phase than for the dissolved phase, regardless of the event, the depth or the intensity of the rainfall. The latter appear to be consistent with a more efficient scavenging of coarse particles

(Al, Fe and Si), predominantly in the particulate phase (Figure 5), compared to the secondary submicronic aerosols (SNA) that make up a large proportion of the dissolved phase (Figure 5), as previously observed in the literature (Kasahara et al., 1996; Ma, 2006). However, the study of the elemental DF suggests that it also varies strongly depending on the element, even when they have a similar predominant phase and similar size within an event (e.g., see Supplement S5, S vs. Cl predominantly in the dissolved phase [>80%] and Ti vs. Cr mainly in particulate

phase [>80%]).

Figure 7 illustrates the differences in the mass concentration decrease of the elements between the first and the last fraction of each rain event. We found that R3, R5 and R6 exhibit lower DF, mostly within a factor 5 for R3 and R3, and within a factor 2 for R5 (Figure 7, Supplement S6). In contrast, DF of R1, R2, R4, R7 (except Ni) and R8 were higher, mainly greater than 5, depending on the element and the event. The lower DFs were observed for the

events characterized by the lower amount and intensity of precipitation (R3, R5 and R6), and therefore a lower efficiency to scavenge the atmosphere. In addition, R5 and R6 were characterized by a long-range transport of mineral aerosols at altitude, which also explains the higher concentrations observed at the end of the event due to the additional contribution of these particles. This observation emphasizes the fact that events can be characterized by local sources (and hence different initial atmospheric concentrations) and cloud inputs that differ between

elements.





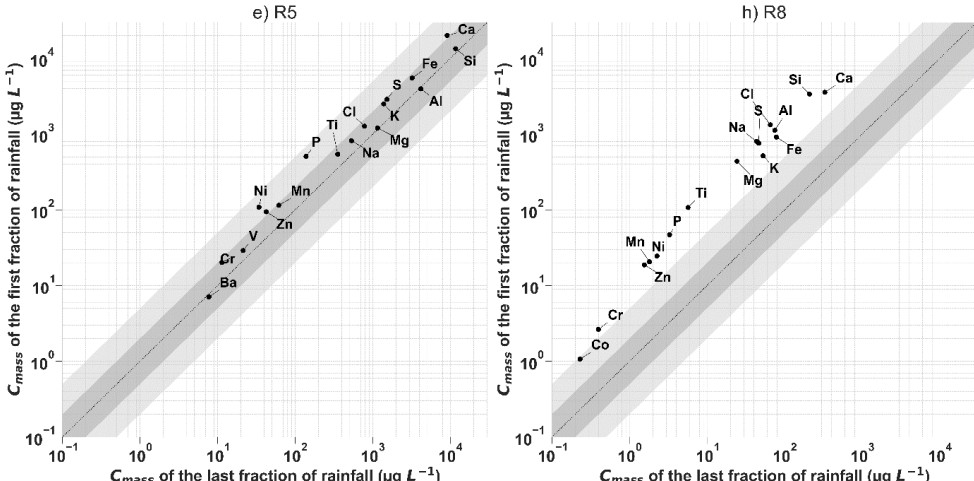

**Figure 7. Element mass concentration in the first fraction of rainfall as a function of elemental mass concentration in the last fraction of (a) R5 and (b) R8. The dotted line represents a 1:1 fit, while the dark and light gray envelopes represent DF within a factors 2 and 5 deviation, respectively.**

Sequential sampling enabled the observation of various patterns of concentration evolution during rainfall events. Some events were characterized by a continuous decrease in mass concentrations throughout the rainfall, ultimately reaching a lower and constant level in the final fractions regardless of the phase nor the chemical species (R1, R7). This kind of evolution is commonly found in the literature, with a high-decreasing trend in the first 1 to 3 mm, until reaching a constant level until the end of the rainfall, for both dissolved and particulate phases (e.g.,

Jaffrezo et al., 1990; Kasahara et al., 1996). In contrast, although lower and constant levels were reached at the end of rainfall, other events exhibited punctual increases or stabilization of the concentrations of both phases during the rainfall (R4, R5 and R8), while the rest of the events showed only punctual increases of the dissolved phase (R2, R3 and R6).

As an illustration, Figure 8 shows the evolution of atmospheric concentrations ($PM_{10}$ and $PM_{2.5}$) with time, the

evolution of mass concentrations of dissolved and particulate phases, rainfall intensity and droplet concentrations (i.e., the number of droplets measured by the disdrometer divided by the unit of volume of the collected rain fraction) during R6 and R8 events. It has been observed that atmospheric concentrations evolve differently according to particle size classes ($PM_{2.5}$ vs. $PM_{2.5-10}$) and rainfall phases. Generally, precipitation is associated with a decrease in atmospheric concentrations during rainfall (Table 1), except for event R6 (Figure 8 f). However, an

increase in concentrations of the coarse aerosol fractions ($PM_{2.5-10}$) is observed quite systematically as rainfall intensities decrease (<0.5 mm h$^{-1}$), especially for events R2, R4, and R8 (between 4:00 and 5:00) as shown in Figure 8a.

Increase of wet deposition concentrations during rainfall has been previously observed by some authors (e.g., Karşı et al., 2018). Here, since the latter were systematically correlated with a decrease in precipitation intensity (Figure

8 d, i) and an increase in droplet concentration (Figure 8 e, j), there could be either due to 1) an effect of "over-concentration" of the falling raindrops or a release of aerosols due to their evaporation (Huff and Stout, 1964;



Baechmann et al., 1996a, b; Gong et al., 2011), or 2) a local emission phenomenon (Karşı et al., 2018). The high temporal resolution of the sampling, and hence the determination of the chemical composition of the dissolved and particulate phases, allows identifying more accurately the cause of these concentration increases.

For rainfall events R4 and R8 (Figure 8 a-e), increases in concentration during the rain are observed for both the particulate and dissolved phases, and are associated with higher precipitation in altitudes than at the surface according to the ceilometer measurements (Figure 5). Partial evaporation of raindrops as they fall could thus reduce their diameter and concentrate them. Assuming that only the water evaporates and not the chemical species contained in the raindrops, the amount of initial material removed by the droplets, expressed in terms of their

volume, is greater (Baechmann et al., 1996b). On the contrary, if the evaporation of the droplets is complete as they fall, this has the effect of releasing aerosols into the atmosphere, thereby increasing atmospheric concentrations (Huff and Stout, 1964; Gong et al., 2011). Therefore, this can increase the concentrations of the following raindrops by capturing released aerosols as they fall.

For R6, there is also an increase in concentrations during rainfall, but only for some species (Figure 8 f-j). $NO_3^-$
and $NH_4^+$ concentrations increase by a factor of 4 to 5, while dissolved Zn and Cu concentrations increase by a factor of 5 to 16 (included in "others"). The increase in $NO_3^-$ and $NH_4^+$ concentrations in precipitation may be due to an increase in local emissions. Indeed, low precipitation rates and a very low boundary layer height (cloud base around 200 m) are observed between 7:00 and 9:00 a.m., a period when road traffic is important and close to the monitoring site. In addition, the NOx concentrations measured at the LISA air quality station also show increases

of more than a factor of 5 over the same time steps. As Zn and Cu are tracers of automotive activity (Thorpe and Harrison, 2008; Bukowiecki et al., 2009; Pant and Harrison, 2013), this supports the hypothesis of the influence of local emissions (road traffic) on the increase in rainfall concentrations throughout the event. R6 is therefore a good case study to illustrate the combined influence of changing meteorological parameters and local sources on the evolution of deposition concentrations during a rain event.



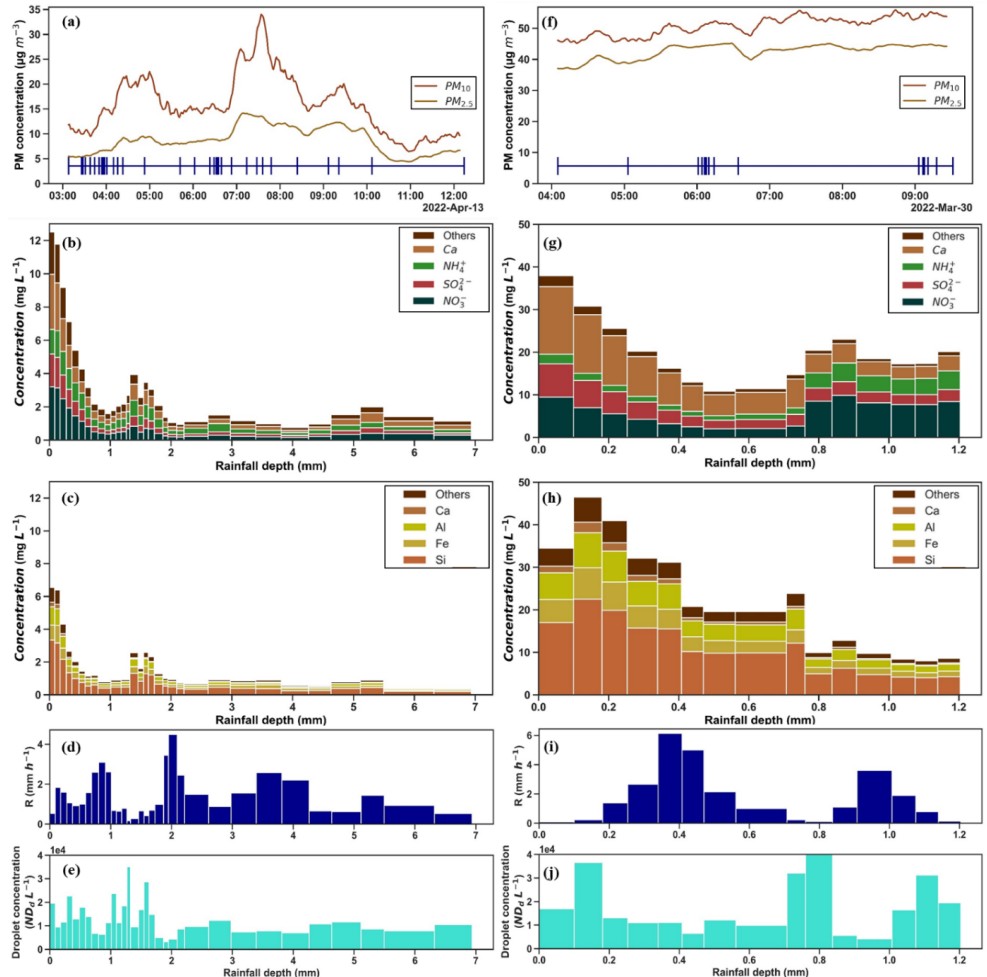

**Figure 8. R8 (a-e) and R6 (f-j) case studies. Evolution of PM$_{10}$ and PM$_{2.5}$ concentrations (µg m$^{-3}$; a and f) with time. The different sampling periods for each rain fraction are indicated by the intervals in blue (a and f). Evolution of dissolved mass concentration (mg L$^{-1}$; b and g), particulate mass concentrations (mg L$^{-1}$; c and h), rainfall intensity (R in mm h$^{-1}$; d and i) and droplet concentration (ND$_d$ L$^{-1}$; e and j) throughout rain events.**

Although the intra-event evolution differs among the case studies, mass concentrations of elements generally decrease rapidly between the beginning and end of each rain event. However, some phases of increasing mass concentrations have been identified during certain events. In order to explain these phenomena, it is necessary to document the precipitation characteristics, atmospheric dynamics, and surface PM$_{10}$ and PM$_{2.5}$ content throughout the rainfall event. We have proposed two hypotheses to explain these punctual increases, namely the over-concentration of rain droplets due to partial or total evaporation, and the effect of local sources.





### 4.3. Contribution of in cloud and below cloud scavenging

Based on the criteria explained Sect. 2.4, we selected R1, R2, R4, and R8 events to discuss the relative contributions of ICS and BCS. The $ICS_C$ of chemical species analyzed in the selected rains are presented in Table 3. We observe different ICS contributions within an event for different chemical species, as well as different 515 $ICS_C$ for the same chemical species between events.

**Table 3. Relative ICS contribution ($ICS_C$) for R1, R2, R4 and R8 events.**

| Chemical species | $ICS_C$ (%) | | | |
|:---:|:---:|:---:|:---:|:---:|
| | **R1** | **R2** | **R4** | **R8** |
| $SO_4^{2-}$ | **62** | 48 | 23 | **58** |
| $NO_3^-$ | 35 | **55** | 27 | **57** |
| $NH_4^+$ | 45 | 40 | 24 | **65** |
| Al | 44 | 38 | 20 | **62** |
| Ba | 37 | **50** | 26 | **68** |
| Ca | 21 | 35 | 16 | **64** |
| Cl | 36 | **88** | 20 | 49 |
| Cr | 44 | **67** | 30 | **75** |
| Fe | 37 | 48 | 26 | **70** |
| K | **67** | 41 | 26 | **70** |
| Mg | 33 | 33 | 18 | **57** |
| Mn | 36 | 41 | 19 | **71** |
| Na | 32 | **85** | 17 | **53** |
| P | 24 | 30 | 17 | **57** |
| Pb | **82** | 37 | 18 | **71** |
| Si | 48 | 31 | 18 | **61** |
| Sr | 21 | 33 | 15 | **60** |
| Ti | 42 | 29 | 17 | **69** |
| V | 37 | **68** | **59** | 37 |
| Zn | **59** | 33 | 18 | **67** |
| **Average ± std** | 42 ± 15 | 47 ± 17 | 23 ± 9 | **62 ± 9** |

For R1, R2 and R4 anthropogenic events, the elements of crustal origin found in the coarse fraction of aerosols (Al, Si, Fe, Ti, Ca, Mg, Sr) are mainly deposited via BCS mechanism. This is consistent with previous *in situ* studies that have shown that the BCS mechanism accounts for 52 – 99% of calcium wet deposition in urban 520 environments (Ge et al., 2016; Karşı et al., 2018; Ge et al., 2021; Berberler et al., 2022). In contrast, deposition of crustal elements in mineral and anthropogenic event R8 are mainly deposited via ICS mechanism (57 – 75%).

With the exception of V (37%) and Cl (49%), all chemical species observed in R8 are mainly deposited through ICS mechanism (57 – 75%). The characteristics of R8 suggest two possible reasons explaining ICS prevalence: 1)





since the precipitation accumulation is higher, the contribution of the rainout mechanism, due to atmospheric
column depletion below the cloud, is also expected to be higher (Ge et al., 2021); 2) the event is characterized by
a long-distance transport of mineral dust, which explains a more pronounced contribution of the ICS for crustal
elements.

For anthropogenic events, Mn and $NH_4^+$ was found to be mainly deposited by the BCS mechanism, between 55
and 87%. This corresponds to a similar range of values reported for $NH_4^+$ in other urban environments in Austria
(65%), Turkey (60 – 95%) and China (47 – 84%) (Xu et al., 2017; Karşı et al., 2018; Berberler et al., 2022;
Monteiro et al., 2021; Ge et al., 2021). In the literature, large variations are found for the contribution of the BCS
mechanism of sulfate and nitrate in urban environments, with values ranging from 50 to 98% (Ge et al., 2016; Xu
et al., 2017; Karşı et al., 2018; Ge et al., 2021; Monteiro et al., 2021; Berberler et al., 2022), and even down to
16% for sulfate (Aikawa et al., 2014). Here, we found $BCS_C$ of sulfate and nitrate ranging from 38 to 77%,
depending on the events. Few chemical species show a predominance of ICS mechanism in the wet deposition of
anthropogenic events that could be due to season explaining the difference in local sources (such as oil and wood
heating systems for $SO_4^{2-}$, Zn), gas scavenging contributions (with nitrate being mainly gaseous in summer and
particulates in winter) (Audoux et al., 2023), or long distance transport. Thus, this explains higher $ICS_C$ for $SO_4^{2-}$
and Zn in R1 in comparison with R2 and R4. The higher $ICS_C$ obtained for Na and Cl for R2 may be explained by
the origin of air masses coming from the Atlantic Ocean. Nevertheless, anthropogenic events are in average found
to be primarily controlled by the BCS mechanism (53 – 77%).

On the basis of the 4 events for which the $ICS_C$ and $BCS_C$ mechanisms were calculated, we cannot conclude on
the influencing factors. Indeed, for the events not characterized by long-range dust transport (i.e., R1, R2 and R4),
no decrease in the average $BCS_C$ with increasing rainfall depth is observed, in contrast to the findings of Ge et al.
(2021). This may be due to a difference in precipitation parameters (i.e., intensity, drop size and cloud base height)
and $PM_{10}$ concentrations (Table 1). Indeed, R1 is characterized by twice the $PM_{10}$ concentrations but 2 to 4 times
lower precipitation depth. In addition, the cloud base height is higher for R4 compared to R2. This could be a
reason why the $BCS_C$ is higher even though the amount of precipitation is higher and the $PM_{10}$ concentration is
lower.

Numerical studies have highlighted the importance of not only cloud height but also cloud thickness in the relative
contribution of BCS and ICS (Kim et al., 2021; Migliavacca et al., 2010; Wiegand et al., 2011). Therefore, cloud
thickness measurements should be planned to better understand the scavenging process and its contribution to the
total wet deposition.

**5. Conclusion**

Measurement campaign has been done in the Paris region to monitor the evolution of chemical composition of wet
deposition with time during rainfall events. The collected rainfall events illustrate contrasting situations in terms
of meteorological conditions (rainfall depth and intensity), atmospheric dynamics (cloud base height between 200
and 2500 m), as well as different atmospheric $PM_{10}$ concentrations ranging from 10 to more than 60 µg m$^{-3}$,
characterized by the urban environment of the study site, but also by mineral dust intrusions from the Sahara.



Using additional measurements, three categories of events were identified according to the origin of the aerosols found in the rain: "anthropogenic" (R1 to R4), "anthropogenic and mineral-dust" (R7 and R8) and "mineral-dust" rainfalls (R5 and R6).

Although total (dissolved + particulate) wet deposition fluxes are not directly related to atmospheric concentrations, there is a link between initial wet deposition mass concentrations, atmospheric $PM_{10}$ concentrations and the $PM_{2.5}/PM_{10}$ ratio. The comparison of the chemical composition of $PM_{10}$ aerosols prior to the rainfall event with the first fraction of rain illustrates the link between surface aerosol and wet deposition content. It allowed us to calculate washout ratios (WR) using wet deposition concentrations at the very beginning of the rainfall, before the dilution effect occurs. WR varied for different elements within and between events, and there was a consistent increasing trend with increasing rainfall.

Our study illustrates the variability of both the mass concentrations and the chemical composition of the particulate and dissolved phases. For the different rains sampled, we observe a rapid decrease in mass concentrations as the rain progresses. The decrease is more pronounced for the particulate fraction (up to a factor of 50) than for the dissolved fraction (up to a factor of 33), regardless of the event, the depth or the intensity of the rainfall. Nevertheless, by studying different case studies, the detailed monitoring of the intra-event evolution of the chemical composition of the wet deposition has made it possible to highlight the predominant role of meteorological parameters, as well as the presence of local sources, in the contrasting evolution of the mass concentrations observed during precipitation.

We estimate the contributions of the in cloud scavenging (ICS) and below cloud scavenging (BCS) mechanisms for some rainfall events (R1, R2, R4 and R8). The results show a significant contribution of both mechanisms, with a higher contribution of the BCS mechanism for rainfall events characterized by a larger anthropogenic contribution and local sources (R1, R2 and R4). However, the contributions of scavenging mechanisms are as variable from one chemical species to another as they are from one rainfall to another, depending on their specific sources, atmospheric dynamic and meteorological conditions. The anthropogenic and mineral event (R8), characterized by long-distance transport of mineral dust, shows a predominant contribution of the ICS mechanism. This illustrates that there can be contrasting situations on the same study site.

These results highlight the importance of understanding the physical and chemical processes involved in the transfer of aerosols from the atmosphere to the precipitation in order to better assess the impact of aerosol particles pollution on the environment.

**Acknowledgment**

This work is performed in the framework of the research programs DATSHA supported by the French national program LEFE (Les Enveloppes Fluides et Environnement) and Foundation Air Liquide, and was also supported by LISA, UPC, UPEC, UMR CNRS 7583 via its internal project call. Some of the analyses (CI, XRF) presented were performed with the instruments of the PRAMMICS platform OSU-EFLUVE UMS 3563.



**Competing interests**

The authors declare that they have no known competing financial interests or personal relationships that could have appeared to influence the work reported in this paper.

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
