# Peer review of "Intra-event evolution of elemental and ionic concentrations in wet deposition in an urban environment"

_EGUsphere, 2023_

## Referee Comment (RC2)

Referee comment on "Intra-event evolution of elemental and ionic concentrations in wet deposition in an urban environment" by Thomas Audoux et al.

This study tried to estimate the mechanism implied in the wet deposition through monitoring the evolution of the chemical composition of wet deposition during rainfall events in Paris region. They found the intra-event observation of each precipitation is useful to reveal the predominant role that affect wet deposition. Overall, this is a nice piece of paper with clear objectives and methods. Before considering publication in ACP, major revisions should be made. Some comments and suggestions are listed as follows:

Major comments:

1. In this paper, the authors collected eight rainfall events. However, only 4 events can be used to discuss the scavenging mechanism. They also notice that the scavenging mechanism varied case by case and cannot conclude as a general conclusion due to little cases. The authors are encouraged to add some discussion on how to improve the "successful monitoring rate" in the future.

2. The discussion section should be reorganized, which now seemed a little messy. For example, section 4.2 and 4.3 can be improved and concluded several findings.

3. In calculated WR, how about the impacts of air pollutants transport on it? They also noticed several cases were influenced by intrusion of mineral dust from northern Africa. They should compare the WR case by case and make a conclusion. Besides, in the first fraction of rainfall, the wind-swept effects should also be considered.

Specific comments:

1. P4, L131. Please accurate describe at which fraction or time the sampling is stopped.

2. P6, L200 and L225. WR was calculated at R2, R3 and R8. However, mechanism was choose as R1, R2, R3 and R8. It is confusing.

3. P7, L208. Please clarify the details on "once a constant level is reached", especially in quantified criterion.

4. P8, L255. "do not seem to be correlated with rainfall depth nor rainfall rate (Table 1)". It seems doesn't make sense. The total rainfall depth should be correlated with the total wet deposition fluxes.

---

## Author Comment (AC1)

We thank the reviewers for their suggestions on how to improve our paper. Responses to each comment are shown in blue.

**RC1**

**This study provides a very useful data set on precipitation scavenging of atmospheric aerosols, which, if properly analyzed, can generate quite some useful information for improving our understanding on this topic. The manuscript covers too wide a scope (e.g., source apportionment of PM in addition to precipitation scavenging) and has too detailed description of each single event, ending up lacking in-depth analysis of key factors/parameters dominating the scavenging process. I have the following comments that may hopefully help improve the presentation quality of this manuscript.**

**Specific comments:**

**Abstract needs a major rewriting. It only describes what has been done, but lacks of a good summary of major findings, especially some quantitative statements, such as the relative contributions of ICS and BSC to the total wet deposition, the range of WR values for some major chemical species investigated here.**

The abstract has been rewritten to emphasize major findings and quantitative results, L11 – L29:

"A measurement campaign was conducted in the Paris region, focusing on the evolution of chemical composition of wet deposition during rainfall events from sequential sampling. A total of eight rain events were documented and characterized by varying meteorological conditions, atmospheric dynamics, and aerosol particle concentrations representative of urban conditions and influenced by long-range mineral dust transport. The intra-event evolution of the chemical composition of wet deposition revealed the predominant role of meteorological parameters and local sources in the observed mass concentration variability. From selected case studies, the washout ratios (WR) and scavenging coefficients were quantified by conducting simultaneous measurements of aerosol particle composition and wet deposition. The results highlighted a variability of the WR and scavenging coefficients depending on the rainfall rate and on the chemical specie. Scavenging coefficients estimated from WR ranged from $5.4 \times 10^{-8}$ to $1.1 \times 10^{-5}$ s$^{-1}$ for chemical elements, and are within the range of values reported in the literature for 0.2-2 μm particles diameter. Our results pointed out that scavenging coefficient increases with rainfall rate according to a power law, as previously shown in the literature, indicating a stronger removal of particles from the atmosphere with greater precipitation intensity. Quantitative analysis of the data allowed us to estimate the relative contributions of in-cloud scavenging (ICS) for selected rain events. The ICS relative contributions ranged on average from 23% to 62% depending on the rain events, and varied according to the chemical species within the same rain event. This highlights the variability and complexity of the wet deposition process and the influence of specific factors on the contribution of ICS, such as aerosol particle size and hygroscopicity. Overall, this study highlights the variability of wet deposition and its chemical composition, and the need to consider the specificities of each event to fully understand the underlying mechanisms."

**Introduction needs a revision to better describe the major goals of this study and related background information (current knowledge). Very basic knowledge should only be briefly mentioned. For example, a large portion of materials in the first three paragraphs is not really needed. While the three objectives (lines 86-91) listed in this section are somewhat important, it would be even more important to quantify the scavenging rate under different meteorological and chemical conditions, and this should be listed as a major goal too.**

As suggested by the reviewer, the introduction has been shortened to get to the point more quickly.

Moreover, we added details on the importance of determining WR.

L59 – L72: "It can also be used to estimate wet deposition fluxes given air concentrations and precipitation rates (Duce et al., 1991; Mamun et al., 2022). However, Cheng et al. (2021) have highlighted in their literature review the scarcity of particulate element WR data due to the limited co-located measurements of elements in precipitation and aerosol particles. […] Therefore, the determination of both WR and scavenging coefficient appears to be very useful for future wet deposition studies."

We also emphasized another study objective to emphasize the importance of determining WR.

L94– L-98: "It has two objectives: (1) to document the intra-event evolution of ionic and elemental composition of dissolved and particulate phase species in wet deposition for contrasted rain events and (2) to discuss the parameters influencing wet deposition chemistry through the quantification of washout ratios and scavenging coefficients and the estimation of the relative contribution of BCS and ICS mechanisms in the wet deposition."

**For example, WR values for the investigated chemical species would be very useful for future wet deposition studies, as demonstrated in Mamun et al. (2022, JGR Atmospheres, 127, e2021JD035787) for elements and Cheng and Zhang (2017, ACP, 17, 4711-4730) for ions. It would be even better if WR values for PM2.5 (fine) and PM2.5-10 (coarse) were calculated separately and summarized in Abstract and/or conclusion section.**

We were unable to calculate the washout ratios (WR) for fine and coarse particles due to the absence of information regarding the elemental composition of the fine (PM2.5) and coarse (PM2.5-10) fractions in the aerosol, which was not included in our conducted measurements. However, drawing inspiration from Cheng and Zhang's study (2017), we could potentially calculate WR associated with either the coarse or fine particulates by utilizing the WR values of elements linked to coarse or fine particles. Nevertheless, while elements associated with coarse particles, such as Ca (WR between 2 500 and 9 800) exhibit higher WR than element associated to fine particles (such as Zn, WR between 1 000 and 3 800, this pattern does not hold true for all cases.

For instance, Al and Si (insoluble elements associated with coarse particles) display WR values similar to those of S (soluble element associated with fine particles). This shows that particle size is yet an important parameter but cannot be taken into account solely and that scavenging processes depends on multiple parameters.

L460 – L484: "In all the cases, the values of WR are higher than the values previously estimated, in agreement with the dilution effect on the WR available in the literature. Indeed, by taking into account the first fraction of the rainfall, the calculation minimizes the influence of the ICS contribution as opposed to the WR values considering the entire event. The differences observed between elements have been discussed in the literature as being due to differences in aerosol characteristics such as size, hygroscopicity or the additional contribution of gas phase scavenging (e.g., Jaffrezo and Colin, 1988; Cheng et al., 2021; Kasper-Giebl et al., 1999; Cheng and Zhang, 2017). Cheng et al. (2021) emphasized the predominant role of particule size distribution on the WR. Indeed, the elements associated with the coarse mode ($PM_{2.5-10}$) present the largest WR, except Si and Fe, while the elements that are dominant in the fine particles ($PM_{2.5}$) had lower WR. Even if we have no information on the size distribution of aerosol chemical composition, the EF shows that the elements associated with coarse mode by Cheng et al. (2021) are from dust origin, and those associated with fine mode (e.g., S, Zn) are of anthropogenic origin, in our samples. Our results are consistent with these observations: while elements linked to coarse particles, such as calcium (WR ranging between 2 500 and 9 800), exhibit higher WR values compared to those associated with fine particles, such as zinc (WR ranging between 1 000 and 3 800). However, as highlighted in the review of Cheng et al. (2021), some elements found primarily in the coarse mode, such as Fe (WR = 3 800), exhibit similar WR value to elements associated with fine particles (e.g., Zn) as illustrated in the event R3.

However, our study revealed a significant variation between different events for the same chemical species. Interestingly, for each element, except S, this variability consistently follows a decreasing trend in WR with increasing pre-rain $PM_{2.5}/PM_{10}$ fraction (Table 1). In addition, we observed an increasing trend in WR with higher rainfall rates. For instance, WR of Ca increase from 2 500 to 9 800 when rainfall rate increases from 0.5 to 1.2 mm $h^{-1}$. This shows that the particle size distribution is probably not the major factor acting on particle below-cloud wet scavenging. These results are particularly noteworthy because they represent the first instance of WR measurements unaffected by the dilution effect."

**Line 255: Such a conclusion/finding is likely because of the very small dataset, in which the impacts of several dominant parameters (rain rate and PM concentration) on total wet deposition cancelled each other, resulting poor correlations between wet deposition flux and these parameters. It would be more useful to investigate WR value under different precipitation rate or PM concentration and generate some useful conclusions.**

Wet deposition fluxes are known to not be only correlated with rainfall depth nor rainfall rate, since it depends on the simultaneous presence of aerosols in the atmosphere and the precipitation that reaches the surface of the Earth. However, the efficiency of aerosol scavenging is indeed influenced by rain rate.

Our finding supports the fact that the total wet deposition fluxes (to be understood here as the wet deposition flux of an entire event) are not correlated with rainfall depth nor rain rate. However, for similar aerosol content, higher rainfall depth can lead to higher wet deposition fluxes as we observe if we consider only the R1, R2, R3, R4 and R8 events. Although here we have a set of 8 rain events, that's why we don't show statistics, but describe the behavior of the data we observe for different situations.

Regarding the discussion of WR under different precipitation rates or PM concentration, we discussed WR values in section 4.1.

Paragraphs have been rewritten L460 – L484 (see response to previous remarks).

**Line 260 and below: The precipitation scavenging data presented in this study would be more useful for investigating precipitation scavenging of atmospheric aerosols, rather than for source apportionment analysis. Discussing air mass trajectory and potential sources of PM add little new knowledge to the topic of precipitation scavenging. Besides, there are numerous studies on PM source apportionment in literature, but very limited studies on precipitation scavenging of atmospheric aerosols using sequential sampling approach. The authors are encouraged to focus more on aerosol scavenging and generate new knowledge that can be used by the scientific community for future wet deposition studies. It seems to me that the data collected in this study can also be used for generating another important parameter, scavenging coefficient, that is typically used in chemical transport models (see Wang et al., 2010, ACP, 10, 5685-5705).**

Description of air mass trajectory, disdrometer and ceilometer measurements have been removed from the result part to focus more on aerosol scavenging and wet deposition. However, classification of case study has been kept (but shortened) because it is important regarding the understanding of intra-event behavior (section 4.1) and scavenging mechanism discussions (section 4.3).

Scavenging coefficient have been calculated and added in the discussion.

L65 – L72: "Another approach is to calculate the scavenging coefficient, which is commonly used in global chemical transport models to represent the below cloud scavenging of particles by rain (Ge et al., 2021b; Colette et al., 2017). Theoretical studies have primarily focused on determining the particle collection efficiency of raindrops as they fall, while certain numerical, laboratory, and field studies have developed semi-empirical parameterizations (Wang et al., 2014; Dépée et al., 2020; Laakso et al., 2003; Slinn, 1977). However, a gap remains between field measurements, theoretical and semi-empirical

parameterizations (Wang et al., 2010, 2011). Therefore, the determination of both WR and scavenging coefficient appears to be very useful for future wet deposition studies."

L211 – L218:
"2.4.2 Scavenging coefficient
We can determine the scavenging coefficient ($\Lambda$, s-1) of elements using field measurements and based on the estimation of their washout ratios, as previously done in the literature for sulfate, nitrate and ammonium (Okita et al., 1996; Xu et al., 2019; Andronache, 2004; Yamagata et al., 2009). Indeed, by assuming a uniformly mixed atmospheric column below the cloud base, the average scavenging coefficient of elements can be expressed using equation (3), R and H being the rainfall rate (in mm s-1) and the average cloud base height (in m) during the first fraction of rainfall, respectively.

$$\Lambda\ (s^{-1}) = WR \times \frac{R}{H} \tag{3}"$$

L485 – L500:

"Scavenging coefficients ($\Lambda$) can be determined from the WR calculation using equation (3). These estimations are the first available for major and trace metals. Figure 6 illustrates $\Lambda$ of elements as a function of rainfall rate. Our results show that $\Lambda$ increases with rainfall rate according to a power law, as previously shown in the literature (e.g., Xu et al., 2019; Wang et al., 2014). At a rainfall rate of R = 1 mm h$^{-1}$, we obtained $\Lambda$ values, between 0.5 and $1.3 \times 10^{-6}$ s$^{-1}$, with the exception of chlorine. These values fall within the range ($2.6 \times 10^{-7}$ – $1.7 \times 10^{-6}$ s$^{-1}$) documented for radionuclides by Sparmacher et al. (1993) for controlled experiments with similar rainfall rate and aerosol diameters (0.98 and 2.16 µm). Scavenging coefficients evolution with rainfall rate that varies from one element to another, with slopes ranging from 0.5 for sulphur to 2.9 for chlorine. These differences cannot be attributed solely to mass concentration, particle size, or water-soluble fraction of elements. For instance, while elements associated with the same aerosol types, such as Na and Cl or Al, Ti and Si, show similar behavior with rainfall rate, chlorine and sulphur exhibit contrasting trends even though they are both water soluble elements. Similarly, scavenging coefficients for coarse particle (e.g., Al and Si; $1.5 – 8.5 \times 10^{-7}$ s$^{-1}$) are comparable to those for fine particle (Zn and S; $0.9 – 6 \times 10^{-7}$ s$^{-1}$). Aerosol scavenging does not depend on a single parameter, but is governed by the interaction of several parameters including the intrinsic properties of the aerosol (size, solubility) and of the precipitation (intensity, size and number of droplets). Consequently, our results underline the critical role of rainfall rates and aerosol particle properties for the determination of both WR and $\Lambda$.

[Figure]

**Figure 6. Scavenging coefficient ($\Lambda$, s$^{-1}$) as a function of rainfall rate (R, mm h$^{-1}$) for studied elements."**

**Section 4 Discussion: most discussions are descriptive of single events, although a brief comparison with literature was provided. I would like to see more in-depth analysis that can provide some definitive or quantitative statements on the dominant factors controlling aerosol scavenging by precipitation.**

Following the reviewers' comments, the discussion section has been reorganized and rewritten to better discuss the results, with a first section on intra-event evolutions, a second on parameterizations (WR and scavenging coefficient), and a third on ICS and BCS contributions, including the addition of new figures.

We work on specific case studies, documenting as many parameters as possible. The discussion then focuses on individual events to highlight parameters of interest. However, the discussions have been revised to better highlight the findings and dominant factors controlling aerosol scavenging by precipitation. Here are the new paragraphs added to highlight the findings.

**Discussion about washout ratios and scavenging coefficients.**

L483 – L484: "These results are particularly noteworthy because they represent the first instance of WR measurements unaffected by the dilution effect."

L497 – L500: "Aerosol scavenging does not depend on a single parameter, but is governed by the interaction including the intrinsic properties of the aerosol (size, solubility) and of the precipitation (intensity, size and number of droplets). Consequently, our results underline the critical role of rainfall rates and aerosol particle properties for the determination of both WR and $\Lambda$."

L575 – L579: "The implications of these results are substantial, as they emphasize the need to consider rainfall characteristics and aerosol properties for accurate estimations of the scavenging process and its impact on atmospheric deposition. Such efforts will help refine and develop more reliable

parameterizations that can accurately represent scavenging efficiency for a wider range of environmental conditions."

**Discussion about intra-event evolution.**

L425 – L430: "By conducting a comprehensive analysis of precipitation characteristics, atmospheric dynamics, and local influences, we aimed to shed light on the underlying mechanisms responsible for the observed punctual increases in mass concentrations during our study cases. Our results highlight the importance of the droplet size distribution, its evolution as well as the presence of local sources that evolve also during the rain event. Such investigations are essential to unravel the complexities of wet deposition dynamics and deepen our understanding of the intricate interactions between atmospheric particles and wet deposition processes."

**Discussion of ICS & BCS.**

L536 – L549: "Several factors may contribute to these differences in the observed contribution of ICS and BCS between events. One key factor is the variation in meteorological conditions, including intensity, droplet size, and cloud base height, as well as $PM_{10}$ concentrations (Table 1). Numerical studies have highlighted the importance of not only cloud height but also cloud thickness in the relative contribution of BCS and ICS (Kim et al., 2021; Migliavacca et al., 2010; Wiegand et al., 2011). This dependence can be explained by the fact that the higher the cloud height, the greater the volume of air swept by the droplets, and therefore the greater the quantity of aerosols encountered by the precipitating droplets, at equal and homogeneous concentration on the atmospheric column. For example, event R1 has higher $PM_{10}$ concentrations but 2 to 4 times lower rainfall depth compared to other anthropogenic events. In addition, R4 has a higher cloud base height compared to R2, which could affect the $BCS_C$ despite the higher precipitation amount and lower $PM_{10}$ concentration. These variations in meteorological conditions and atmospheric dynamic could influence BCS efficiency as well as aerosol content to be scavenged below the cloud, leading to the observed discrepancies in $BCS_C$ and $ICS_C$ values. Consequently, the complex interactions between meteorological conditions, aerosol properties, local sources and long-range transport can result in different scavenging behaviors for each event, highlighting the challenge and the need of wet deposition studies."

L592 – L595: "To gain a comprehensive understanding of the factors influencing scavenging mechanisms, further investigation is necessary, including a larger data set covering a wider range of meteorological conditions and aerosol characteristics. Such a comprehensive approach will enable a more robust analysis and to confirm and/or identify the dominant factors that drive scavenging during rainfall events."

**Section 5 Conclusion: I have similar comments on this section as I have on the Abstract.**

Conclusion has been rewritten to emphasize major findings and quantitative results

**Minor comments:**

**Line 15: Why use "The latter" when you only have one thing mentioned in the proceeding sentence?**

Sentence have been modified when correcting the abstract.

**Lines 16-18: fix the grammar issue in this sentence.**

Sentence have been modified when correcting the abstract.

**Line 44: Is WR defined here the same as "the scavenging ratio" frequently used in literature? Better clarify this point here.**

This has been added L53 – L54: "One approach is to compute the washout ratio (also called scavenging ratio and hereafter referred to as WR)"

**Line 94-96: better write this way: "The sampling site is located at the air quality station operated by the Interuniversity Laboratory of Atmospheric Systems (LISA), which is inside the University of Paris Est Creteil (UPEC)….."**

This has been modified.

**Line 167: change "scavenging" to "scavenged"**

This has been modified.

**Line 201-203: difficult to understand.**

Sentence has been rewritten.

L220 – L224: "The relative contribution of the ICS mechanism to the measured wet deposition is determined by analyzing the mass concentrations of chemical species measured at the end of rainfall (referred to as $C_{ICS}$). Indeed, due to the scavenging during the initial stages of rainfall, the end of rainfall is characterized by lower PM concentration, which makes the BCS mechanism negligible in terms of wet deposition (e.g., Aikawa and Hiraki, 2009) since the rain composition can be considered representative of the concentrations of droplets in the cloud."

**RC2**

This study tried to estimate the mechanism implied in the wet deposition through monitoring the evolution of the chemical composition of wet deposition during rainfall events in Paris region. They found the intra-event observation of each precipitation is useful to reveal the predominant role that affect wet deposition. Overall, this is a nice piece of paper with clear objectives and methods. Before considering publication in ACP, major revisions should be made. Some comments and suggestions are listed as follows:

**Major comments:**
**1. In this paper, the authors collected eight rainfall events. However, only 4 events can be used to discuss the scavenging mechanism. They also notice that the scavenging mechanism varied case by case and cannot conclude as a general conclusion due to little cases. The authors are encouraged to add some discussion on how to improve the "successful monitoring rate" in the future.**

The challenge with field measurements is that rainfall events as well as its intra-event evolution cannot be accurately predicted. Improving the "successful monitoring rate" of natural events is therefore a daunting task. That's why we are working on specific case studies, documenting as many parameters as possible. While we cannot draw general conclusions from our case studies, our results confirm what has been reported in the literature over the past few years. Namely, that below-cloud scavenging is not negligible regarding wet deposition, despite long-standing claims to the contrary. In addition, our results provide new directions for future research, particularly regarding the effect of droplet size distribution (which has been poorly investigated in the field with a focus on wet deposition) and the effect of cloud base height (which, to the best of our knowledge, has only been investigated through numerical studies).

Sentence have been added in the conclusion to highlight the findings and provide new directions for future research:
L590 – L595: "our findings provide new directions for future research, particularly regarding the effect of droplet size distribution and the effect of cloud base height on wet deposition dynamics.

To gain a comprehensive understanding of the factors influencing scavenging mechanisms, further investigation is necessary, including a larger data set covering a wider range of meteorological conditions and aerosol characteristics. Such a comprehensive approach will enable a more robust analysis and to confirm and/or identify the dominant factors that drive scavenging during rainfall events."

**2. The discussion section should be reorganized, which now seemed a little messy. For example, section 4.2 and 4.3 can be improved and concluded several findings.**

Following the reviewers' comments, the discussion section has been reorganized and rewritten to better discuss the results, with a first section on intra-event evolutions, a second on parameterizations (WR and scavenging coefficient), and a third on ICS and BCS contributions, including the addition of new figures.

Section 4.1 has been divided into two subsections.

Section 4.3 has been modified to better highlight the findings.

**3. In calculated WR, how about the impacts of air pollutants transport on it? They also noticed several cases were influenced by intrusion of mineral dust from northern Africa. They should compare the WR case by case and make a conclusion. Besides, in the first fraction of rainfall, the wind-swept effects should also be considered.**

For the calculation of WR, we specifically choose to consider only the first fractions of the wet deposition event so that the concentration measured in the rainfall is mainly due to BCS mechanism.

Therefore, the impact of intrusion of mineral dust from northern Africa is limited for the transport in altitude. However, the low fraction of PM2.5/PM10 in the pre-rain concentration must be due to the transport of mineral dust in the atmospheric surface. Nevertheless, it is less the fact that it is transported from northern Africa than the fact that it is coarser particles that have an impact on the calculated WR.

Regarding wind-swept effect, the wind velocity during the collection of the first fractions of R2, R3 and R8 are quite low, $2.7 \pm 0.6$ m s$^{-1}$, $3.2 \pm 0.7$ m s$^{-1}$ and $1.5 \pm 0.3$ m s$^{-1}$, respectively. In the literature, some thresholds are often used to select events in order to study scavenging effect and prevent situations influenced by turbulent mixing or meteorology linked effect. For example, variations of wind speed that must be below 2 m s$^{-1}$ during the rain event (e.g., Oduber et al., 2019; Blanco-Alegre et al., 2021). Therefore, we consider that the wind has little effect on the wet deposition during these fractions, and thus little effect on the determination of WR.

Paragraphs have been rewritten.

**Specific comments:**

**1. P4, L131. Please accurate describe at which fraction or time the sampling is stopped.**

Air sampling is stopped immediately after wet deposition collection begins. Sampling is stopped as soon as possible, within a minute after removing the cover from the sequential sampler, while the first fraction is being collected.

Additional details have been added L137 – L139: "Air sampling is done between 15 hours and 24 hours before the start of the rain and is stopped at the beginning of the latter, within one minute after removing the cover from the sequential sampler, while the first fraction is being collected."

**2. P6, L200 and L225. WR was calculated at R2, R3 and R8. However, mechanism was choose as R1, R2, R3 and R8. It is confusing.**

To address this concern, we will clarify in the manuscript that the WR was calculated specifically for the rain events R2, R3, and R8 while the discussion about scavenging mechanism have been possible only for R1, R2, R4 and R8.

To discuss WR, we need information of 1) the concentration of elements found in the first fraction of the rain, which we have for each rain, and 2) the concentration of elements in the aerosol, which we have only for R2, R3, R5 and R8. That is why we could calculate WR only for R2, R3, R5 and R8. In addition to the elemental concentration in both phases, we have filtered out the R5 study case for the WR calculation due to the high-altitude transport of mineral aerosols, making the collected aerosol sample unrepresentative of the air column.

Additional details explaining the removal of R5 case study for the WR calculation have been added to the Materials and Methods section L205 – 210:

"To accurately calculate the WR, it is important to consider the homogeneity of the atmospheric column to ensure the representativeness of surface aerosol measurements. In our study, we observed the presence of a high-altitude aerosol layer using ceilometer measurements (Supplement S2). The atmospheric transport of mineral dust at high altitudes rendered the collected aerosol sample unrepresentative of the scavenged air column. As a result, we excluded the R5 study case from the WR calculation. Therefore, we will focus our discussion on the WR of the element only for R2, R3, and R8."

To discuss mechanism, we have established selection criteria. We did not chose to treat different case studies. According to our selection criteria, our dataset allows us to discuss the scavenging mechanisms for these rain events.

The selection criteria and thus the filtration of the data have been explained in the Materials and Methods section L229 – L236:

"In our case, we selected rainfall events for which the measurements indicated an effective scavenging of the atmospheric column, with a predominant relative contribution of ICS at the end of the event. To select these events, we used the following criteria: 1) the decrease of concentrations measured in the wet deposition, reflecting the evolution of the contribution of the BCS; 2) the decrease of atmospheric concentrations measured using the FIDAS, suggesting a progressive scavenging of the air column under the cloud; and 3) constant concentrations of wet deposition at the end of the event, indicating a steady state between ICS and BCS. From these criteria, the relative contributions of the scavenging mechanisms could be discussed for R1, R2, R4 and R8 case studies."

**3. P7, L208. Please clarify the details on "once a constant level is reached", especially in quantified criterion.**

Paragraph have been reorganized in order to be able to add details on the steady states reached.

L237 – L239 : "We determine $C_{ICS}$, using the VWM of the last fraction of rain, once a steady state is reached at the end of the rainfall for R1 (1.48–2.65 mm), R2 (1.02–1.33 mm for SNA and 0.89–1.33 mm for other elements), R4 (2.21 – 4.42  mm) and R8 (1.87 – 6.94 mm)."

**4. P8, L255. "do not seem to be correlated with rainfall depth nor rainfall rate (Table 1)". It seems doesn't make sense. The total rainfall depth should be correlated with the total wet deposition fluxes.**

No, the amount of precipitation is not correlated with the total wet deposition fluxes; wet deposition does not depend solely on the amount of precipitation or its intensity. Wet deposition requires the simultaneous presence of atmospheric aerosols and precipitation. High rainfall depth does not necessarily mean high wet deposition fluxes.

For example, if we look at R5 event, which is characterized by a total rainfall depth of 0.9 mm, it represents a total deposition flux more than 2 times higher than R8 event (106 g m$^{-2}$ vs. 48 g m$^{-2}$), which is characterized by a total precipitation 8 times higher (7 mm).

However, for similar aerosol content, higher rainfall depth can lead to higher wet deposition fluxes as we observe if we consider only the R1, R2, R3, R4 and R8 events.

However, the original statement was not intended to be general, but to describe our case studies. The sentence has been modified to avoid confusion.

L291 – L295: "Total wet deposition fluxes in our case studies are ranging from 11 to 107 g m$^{-2}$ and are not correlated with rainfall depth nor rainfall rate (Table 1). Indeed, higher wet deposition fluxes are observed for rainfall events (R5 and R6) associated with low rainfall depth but higher pre-rain PM10 concentration.  However, events characterized by a similar surface PM10 mass concentration (R1, R2, R3, R4 and R8) exhibit total wet deposition fluxes that vary over a factor 4."